# PRLS-RFF: Physically Consistent Representation Learning with Self-Supervised Pre-training for RF Fingerprinting

## Abstract

Under domain shifts and open-world conditions, reliably re-identifying radio frequency (RF) devices is essential for wireless security. As a hardware-rooted physical-layer signature, the RF fingerprint has been widely used for device re-identification. However, supervised RF fingerprint identification (RFFI) models often overfit acquisition artifacts and rely on extensive supervision, leading to sharp cross-domain performance drops and weak open-world behavior. To address these limitations, we introduce PRLS-RFF, which targets physically consistent RF fingerprint representations. We design a dual-stream Mamba-based backbone with physics-consistent, multi-view perturbations to encode representation-level invariances. Additionally, to better capture the structure of RF fingerprints (RFFs) across transient and steady-state regimes, the backbone fuses time-domain and time–frequency features using efficient long-context modeling. As a result, the learned representations are domain-robust, which can support reliable open-set recognition. To validate its robustness, we conduct extensive experiments on several public datasets and observe the performance surpassing state-of-the-art models in both cross-domain identification and open-set recognition tasks.

## 1 Introduction

With wireless communications proliferating, device re-identification (re-ID) has become a foundational security primitive across IoT, edge, and low-altitude drone networks (Zhao et al., 2024b; Xu et al., 2016). Precise device-level re-ID underpins access control, auditing, and incident response, safeguarding the reliability and trust of wireless infrastructures (Jawne et al., 2025).

Traditional wireless device recognition typically relies on Pulse Descriptor Words (PDWs), traffic-level statistics, and other conventional descriptors, which are brittle to channel dynamics, hardware variability, and environmental interference (Li et al., 2022; Marchal et al., 2019). In contrast, RF fingerprints (RFFs), unintentional physical artifacts embedded in baseband waveforms, originate in the physical layer and remain tightly coupled to individual transmitters (Li et al., 2022; Kong & Chen, 2025). Extensive prior work shows that RFFs provide highly discriminative, device-specific cues that improve re-identification robustness (Yang et al., 2024).

Having established RF fingerprint as physical-layer signatures, the central challenge is learning representations that stay robust to channel changes and acquisition artifacts. Supervised classifiers trained on labeled data degrade under domain shifts. They also struggle in open-world settings with unseen devices (Yang et al., 2024). Their heavy dependence on labels also fosters shortcut learning, overfitting to spurious channel or collection artifacts rather than device-intrinsic fingerprints (Zhao et al., 2024b). These limitations motivate the learning paradigms that (i) reduce the dependence on labels, (ii) explicitly encode physical-layer invariances, and (iii) generalize across domains while detecting unknown devices (Han et al., 2025; Cai et al., 2024; Zhao et al., 2024a).

To address these challenges, we introduce PRLS-RFF, a dual-stream multi-view self-supervised pretraining framework that couples physics-consistent RF perturbations with multi-view representation learning (Grill et al., 2020; Bardes et al., 2022). By exposing the raw waveform signal to physically grounded variations and enforcing cross-view consistency (Tian et al., 2020), the backbone is driven to encode device-intrinsic fingerprints that remain stable across channels, receivers, and environments. To strengthen the extraction of RF fingerprint cues in both temporal traces and time–frequency structure, we employ a dual-stream encoder that fuses the native waveform with a complementary time–frequency branch. Extensive experiments show consistent gains over strong baselines in cross-domain identification and open-world recognition. The main contributions of this paper are highlighted as follows:

- We propose PRLS-RFF, a dual-stream backbone that fuses time-domain and time–frequency cues and is pretrained via self-supervision for radio frequency fingerprinting.

- We introduce physics-consistent augmentations and a multi-view consistency objective to learn representations that preserve device identity across channel and receiver shifts, enabling consistent classification under changing conditions.

- We introduce selective state-space (Mamba) blocks into the backbone and validate their effectiveness for long-context RF sequence modeling.

- Extensive evaluations on public datasets show consistent gains in cross-domain identification and open-world recognition, with lower annotation needs.

## 2 Related Work

### 2.1 Radio Frequency Fingerprint Identification (RFFI) Methods

Traditional device identification based on protocol and traffic-level features is operationally useful but brittle under channel dynamics and environmental variability (Marchal et al., 2019; Li et al., 2022; Xu et al., 2016). RFFI methods instead exploit subtle but stable hardware-induced artifacts at the physical layer, enabling finer device-level separability (Zhang et al., 2021; Jawne et al., 2025). Supervised RFFI classifiers perform strongly in-distribution, but degrade with receiver, distance, signal-to-noise ratio (SNR), and temporal shifts. Open-world settings with unseen devices further stress purely supervised pipelines (Zhang et al., 2021; Yang et al., 2024). To mitigate these issues, recent efforts advance along three lines: (i) Interpretable and scalable modeling decouples nuisance factors and leverages augmentation or few-shot adaptation (Zhao et al., 2024b;a). Recent work pursues interpretable, scalable modeling with augmentation or few-shot adaptation, but often assumes target-domain access, still needs supervised finetuning, and uses weakly physical augmentations, so shortcut cues persist under shift (Zhao et al., 2024b;a). Open-world and semi-supervised methods also face brittle proxy-tuned thresholds and noise-sensitive pseudo-labeling (Han et al., 2025). We instead shift robustness upstream with self-supervised pretraining on physics-consistent multi-view signals, yielding domain-stable representations with fewer labels.

### 2.2 Backbone Architectures for RF Fingerprint

**1D temporal vs. 2D time–frequency pipelines.** One branch of RFFI backbones operates directly on 1D I/Q sequences with CNN/RNN/LSTM/Transformer variants, trading off local pattern extraction, long-range dependency modeling, and computational cost (Han et al., 2025; Zhao et al., 2024a; Trabelsi et al., 2018). The other branch maps signals to time–frequency representations and applies image-style backbones to capture structured spectral patterns (Cai et al., 2024). While complementary under varying SNR and multipath, single-modality designs face structural limits: 1D models may underutilize stable spectral envelopes and hardware distortion signatures, and 2D scalogram can obscure transient/phase cues and introduce windowing/hyperparameter bias. In practice, these choices leave gaps under domain shifts and encourage shortcut reliance on acquisition artifacts (Zhang et al., 2021; Yang et al., 2024; Zhao et al., 2024b).

**Efficient long-context models.** Transformer architectures offer global receptive fields but incur quadratic cost and memory, complicating long-context training and deployment (Han et al., 2025). Selective state-space models (SSMs), notably Mamba, deliver linear-time sequence modeling with hardware-friendly parallelism while retaining long-range capacity (Gu & Dao, 2024). In RF, RFMamba (Zhang et al., 2025) introduces frequency-aware SSMs for human-centric perception, empirically validating SSM efficacy on long RF sequences. Prior RF SSM studies are largely task-specific and single-stream. We therefore adopt a Mamba-based dual-stream design that fuses a time-domain encoder on raw I/Q with a continuous wavelet transform (CWT) time–frequency encoder into a unified representation.

## 3 Method

We present PRLS-RFF, a physics-consistent dual-stream backbone with stacking Mamba fusion and self-supervised pretraining for multiple views to learn device-intrinsic radio frequency fingerprints (RFFs). The backbone processes a time-domain I/Q stream and a CWT-based time–frequency stream. Their features are fused by stacked shape-preserving Mamba blocks to form a unified representation.

### 3.1 Problem Setup and Notation

RF fingerprints arise from device-specific analog front-end nonidealities at the physical layer and manifest in both transient and steady-state regimes (Soltanieh et al., 2020; Danev et al., 2012; Zhang et al., 2021). To jointly model the transient and steady-state characteristics, we couple a time-domain view that preserves fine-grained transients and phase evolution with a time–frequency view that emphasizes steady-state spectral structure and physical-layer patterns. The time-domain view uses the raw I/Q waveform. For the time–frequency view, we adopt the continuous wavelet transform (CWT) scalogram, which provides scale-dependent time–frequency resolution suited to non-stationary RF signatures (Lin et al., 2020).

Let $x \in \mathbb{C}^T$ be a complex baseband I/Q sequence sampled at the rate $f_s$. To make the CWT setup dataset-agnostic, we normalize $x \in \mathbb{C}^T$ to $[-1, 1]$ and denote the result by $s$. We index by $n \in \{0, \dots, T-1\}$ and write $s[n] \in \mathbb{C}$ for the $n$-th complex sample. For any signal $s$ drawn from the normalized input or its augmented variants, we form two modalities:

$$\text{time-domain:} \qquad r^{\text{TD}} = s \in \mathbb{C}^T, \qquad (1)$$

$$\text{time–frequency:} \qquad r^{\text{TF}} = \text{CWT}(s) \in \mathbb{R}^{F \times U}, \qquad (2)$$

where $\text{CWT}(\cdot)$ denotes the continuous wavelet transform with $F$ and $U$ the index sets of discretized scales and time samples. For a normalized continuous-time signal $s(t)$, the CWT at scale $a$ and shift $b$ is

$$W_s(a, b) = \frac{1}{\sqrt{|a|}} \int_{-\infty}^{\infty} s(t) \, \psi^* \left( \frac{t-b}{a} \right) \, dt, \qquad (3)$$

where $\psi$ is the chosen wavelet function, and $(\cdot)^*$ denotes complex conjugation.

### 3.2 Physics-Consistent Multi-View Augmentations

Real-world RFFI pipelines are impacted by nuisance factors such as channel variability, SNR fluctuations, and analog-chain nonidealities. Supervised models may overfit these shortcuts instead of device-intrinsic cues (Marchal et al., 2019; Zhang et al., 2021). Prior work often uses augmentation as a generic regularizer (e.g., AWGN or simple time slicing) without tying the transforms to a cross-view consistency objective (Han et al., 2025; Cai et al., 2024; Zhao et al., 2024a). In contrast, we construct physics-consistent multi-view (Hendrycks* et al., 2020) signals by applying complex-domain operators that act directly on the I/Q baseband. Unlike common real-valued 1D time-series transforms that ignore the complex structure of I/Q, our four perturbations are standard RF impairments defined in the complex plane. All operators are defined on the normalized complex baseband sequence $s[n] \in \mathbb{C}$ (Sec. 3.1). For clarity, we write $s[n] = I[n] + jQ[n]$ and $x[n] = [\, I[n], \, Q[n] \,]^\top \in \mathbb{R}^2$. Parameters are sampled from bounded, semantics-preserving ranges (listing in Appendix table 11).

**Noise (AWGN).**

$$v_{\text{noi}}[n] \;=\; s[n] \,+\, w[n], \qquad w[n] \sim \mathcal{CN}(0, \sigma^2). \tag{4}$$

Here $w[n]$ is zero-mean circular complex Gaussian noise with variance $\sigma^2$ chosen to realize a target SNR (dB). AWGN is known to improve robustness under low SNR and channel variation (Soltani et al., 2020b; Shen et al., 2023; Comert et al., 2022).

**Phase distortion (phase noise / slow wander).**

$$v_{\text{pha}}[n] \;=\; s[n]\, e^{\mathrm{j}\,\theta[n]}. \tag{5}$$

The process $\theta[n]$ is *slowly varying* (e.g., low-pass noise or a random walk), inducing time-varying constellation rotation while preserving symbol order, and its magnitude is bounded to keep modulation semantics. Phase noise is a device-related impairment and a useful fingerprinting cue (Ali & Fischer, 2019).

**Carrier frequency offset (CFO).**

$$v_{\text{cfo}}[n] \;=\; s[n]\, e^{\mathrm{j}\,2\pi\,\Delta f\, n/f_s}. \tag{6}$$

Here $\Delta f$ (Hz) models oscillator mismatch or motion-induced Doppler. $f_s$ is the sampling rate, and $2\pi\Delta f\, n/f_s$ is the cumulative phase ramp. CFO is hardware-related and has been used as a fingerprint or auxiliary cue (Ali & Fischer, 2019; Zhang et al., 2021).

**I/Q imbalance.**

$$v_{\text{I/Q}}[n] \;=\; \alpha\, s[n] \,+\, \beta\, s^*[n]. \tag{7}$$

The complex coefficients $\alpha, \beta \in \mathbb{C}$ capture I/Q gain and phase mismatch as well as image leakage. A common parameterization uses linear gain ratio $\varepsilon$ and phase error $\varphi$:

$$\alpha \;=\; \tfrac{1}{2}(1+\varepsilon)\, e^{\mathrm{j}\,\varphi/2}, \qquad \beta \;=\; \tfrac{1}{2}(1-\varepsilon)\, e^{-\mathrm{j}\,\varphi/2},$$

so that the image level is controlled by $|\beta/\alpha|$. I/Q imbalance yields stable device signatures and is significant enough that recent work attempts to conceal it (Yang & Li, 2024; Yao et al., 2024).

At each pretraining step we instantiate $\{v_{\text{noi}}, v_{\text{pha}}, v_{\text{cfo}}, v_{\text{I/Q}}\}$, pair each view with the original $s$, and minimize the non-contrastive consistency loss across pairs.

### 3.3 Dual-Stream Encoders and Modal Fusion with Mamba block

Building on Sec. 3.1, we use a dual-stream encoder to capture transient (time-domain) and steady-state (time–frequency) cues (Fig. 1). Both streams are parameterized by stacks of standard selective state-space (Mamba) layers, without any architectural modification, and we adopt this choice after comparing Mamba against Transformer, CNN, and RNN-based long-sequence backbones under the same framework (Appendix A.1).

We use encoders with the same model width for the time-domain and time–frequency branches, since both inputs are deterministic, physics-consistent transforms of the same RF signal and thus offer complementary views of the same hardware-level fingerprint; matching their capacity makes the two representations more comparable and easier to fuse. In the time-domain stream, the normalized I/Q waveform is linearly projected to the model width and processed by a shape-preserving Mamba block. In parallel, the time–frequency stream computes a CWT scalogram, applies a $1 \times 1$ convolution for channel reduction, flattens the result into a token sequence, projects it to the same width, and feeds it to a second Mamba block. The two encoded sequences are then concatenated and passed through a projection layer that reshapes them into a form compatible with stacked Mamba fusion blocks with shared parameters, which encourages the two modalities to live in a common representation space while remaining parameter-efficient. These fusion blocks finally produce the latent RF-fingerprint representation $H_{\text{RFF}}$. Formally, the forward pass comprises two parallel streams.

**Time-domain stream.**

$$H_{\text{TD}} = \mathcal{M}_{\text{TD}}\big(\text{FC}_{\text{TD}}(s)\big), \tag{8}$$

where $s \in \mathbb{C}^T$ is the normalized baseband I/Q, $\text{FC}_{\text{TD}}$ projects to width $d$, and $\mathcal{M}_{\text{TD}}$ is a shape-preserving Mamba block.

**Time–frequency stream.**

$$H_{\text{TF}} = \mathcal{M}_{\text{TF}}\Big(\text{FC}_{\text{TF}}\big(\text{reshape}(\text{Conv}_{1\times1}(S_{\text{TF}}))\big)\Big), \tag{9}$$

where $S_{\text{TF}} = \text{CWT}(s) \in \mathbb{R}^{F \times U \times C_{\text{TF}}}$ ($C_{\text{TF}}$ is the number of time–frequency channels) is the scalogram, $\text{Conv}_{1\times1}$ reduces channels, reshape flattens the grid into a token sequence, $\text{FC}_{\text{TF}}$ projects to width $d$, and $\mathcal{M}_{\text{TF}}$ is shape-preserving.

**Fusion.**

$$Z = \big[\, g_{\text{TD}}(H_{\text{TD}}) \,\|\, g_{\text{TF}}(H_{\text{TF}}) \,\big], \qquad H_{\text{RFF}} = \big(\mathcal{M}_{\text{fuse}}\big)^N\big(\text{FC}_{\text{fuse}}(Z)\big). \tag{10}$$

Here $g_{\text{TD}}$ and $g_{\text{TF}}$ align the stream widths, $[\cdot \| \cdot]$ concatenates along the feature dimension to form $Z$, $\text{FC}_{\text{fuse}}$ projects $Z$ to match the input width required by the Mamba fusion blocks, $(\mathcal{M}_{\text{fuse}})^N$ applies $N$ stacked shape-preserving fusion blocks, and $H_{\text{RFF}}$ is the fused latent RF-fingerprint representation used by downstream heads.

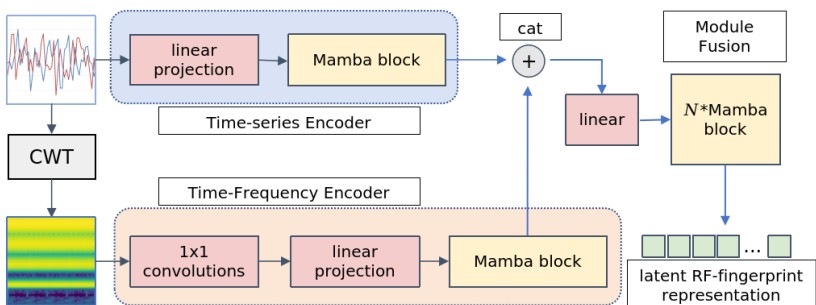

Figure 1: Overview of the dual-stream Mamba backbone.

### 3.4 BACKBONE PRETRAINING WITH SELF-SUPERVISED LOSS

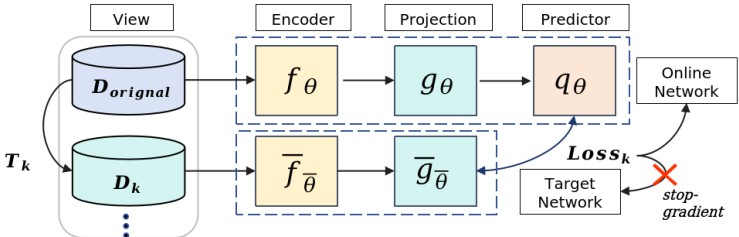

Figure 2: Pretraining with physics-consistent multi-view (sg denotes stop-gradient).

With the dual-stream backbone (Sec. 3.3) and physics-consistent multi-view perturbations (Sec. 3.2) in place, we seek to turn these views into explicit invariance constraints for the backbone. Naively training on augmented data as a generic regularizer can still overfit channel- or receiver-specific shortcuts, and single-network self-reconstruction often risks representation collapse or trivial solutions in the absence of negatives. We therefore adopt a multi-view self-supervised non-contrastive learning framework in which four physics-consistent views are aligned with an original prototype via an EMA-updated online–target architecture, so each signal only interacts with its own views and the learned representation is biased toward hardware-consistent RF fingerprints rather than domain-specific artifacts (see Appendix A.2 for a comparison with a SimCLR-style variant).

As illustrated in Fig. 2, for each $s$ we form four view pairs, and each pair has: the original view $D_{\text{original}}$ and a transformed view $D_k = T_k(s)$ produced by the $k$-th physics-consistent transform $T_k$. We use an online encoder $f_\theta$, projector $g_\theta$, and predictor $q_\theta$, together with a momentum-updated target branch $(\bar{f}_{\bar{\theta}}, \bar{g}_{\bar{\theta}})$ (Grill et al., 2020). The target branch has no predictor and is stop-gradient ($sg$). The target parameters are updated by exponential moving average (EMA):

$$\bar{\theta} \leftarrow m\bar{\theta} + (1-m)\theta, \qquad m \in [0,1). \tag{11}$$

Here, $\theta$ collects the online parameters and $\bar{\theta}$ the target parameters. For any signal $s$, we write

$$p_\theta(s) = q_\theta\big(g_\theta(f_\theta(s))\big), \tag{12}$$

$$z_{\bar{\theta}}(s) = \text{sg}\big(\bar{g}_{\bar{\theta}}(\bar{f}_{\bar{\theta}}(s))\big), \tag{13}$$

and let $\hat{u} = u/\|u\|_2$ denote $\ell_2$ normalization.

Given the normalized original signal $s$ and four augmented views $v_k \in \{v_{\text{noi}}, v_{\text{pha}}, v_{\text{cfo}}, v_{\text{I/Q}}\}$, we symmetrize the alignment around the original (prototype) with forward and backward terms:

$$\mathcal{L}_k^{\rightarrow} = 2 - 2\langle \widehat{p_\theta(s)}, \widehat{z_{\bar{\theta}}(v_k)} \rangle, \qquad \mathcal{L}_k^{\leftarrow} = 2 - 2\langle \widehat{p_\theta(v_k)}, \widehat{z_{\bar{\theta}}(s)} \rangle, \qquad k = 1, \ldots, 4. \tag{14}$$

The total pretraining loss sums all view-to-prototype alignments:

$$\mathcal{L}_{\text{total}} = \sum_{k=1}^{4} \left( \mathcal{L}_k^{\rightarrow} + \mathcal{L}_k^{\leftarrow} \right). \tag{15}$$

The predictor $q(\cdot)$ is used only on the online branch. The target branch is stop-gradient and contains no predictor. Together with EMA in Equation (11), this avoids collapse while leveraging multi-view consistency to extract device-intrinsic RF fingerprint representations.

## 4 EXPERIMENTS

We assess PRLS-RFF primarily through gains on cross-domain identification and open-world recognition after backbone pretraining. For each dataset we first pretrain the backbone for 1000 epochs using our pretraining of Sec. 3. We use a base learning rate of $1 \times 10^{-5}$, the Adam optimizer, and a batch size of 512. The Mamba block in the backbone used in all experiments are implemented as stacks of four standard Mamba layers with state dimension $d_{\text{state}} = 256$ and model width $d_{\text{model}} = 192$. This per-block configuration is selected as a favourable accuracy–efficiency trade-off based on the ablation in Appendix A.1.2.

### 4.1 CROSS-DOMAIN IDENTIFICATION

**Task definition and dataset.** Cross-domain evaluation tests robustness to domain shifts across collection conditions. Labels remain unchanged, but the input distribution varies with channel, SNR, and hardware drift, which often degrades purely supervised models. We use the public UAV RF fingerprint dataset (Soltani et al., 2020a). I/Q signals from seven identical DJI M100 UAVs were collected at four receiver distances (6, 9, 12, 15 ft), with four time-separated acquisition bursts per distance, inducing distance and session shifts. To further probe cross-domain robustness across multiple receivers and weeks-long collection gaps, we additionally evaluate on the ManySig subset of the WiSig dataset (Hanna et al., 2022). ManySig contains I/Q sequences from 6 transmitters observed at 12 receivers, with 1000 length-256 segments per Tx–Rx pair collected on four capture days (03-01, 03-08, 03-15, 03-23).

**Experimental setup and results.** Following Cai et al. (2024), we form two groups of datasets (`data#1` and `data#2`) collected at different distances. Each group is partitioned

Table 1: Cross-domain identification on the UAV RF fingerprint dataset: top-1 accuracy (%) on `data#1` and `data#2`, following the evaluation protocol of Cai et al. (2024).

| Method | data#1 | data#2 |
|---|---|---|
| CNN-LSTM (Cai et al., 2022) | 87.15% | 92.91% |
| GRU (Shen et al., 2022) | 87.02% | 91.72% |
| Baseline (Cai et al., 2024) | 93.05% | 98.90% |
| **PRLS-RFF** | **98.14%** | **98.97%** |

Table 2: Few-shot cross-domain Top-1 (%) on the UAV RF fingerprint dataset. D–T denotes distance–burst. $K$ is shots per class. Baseline follows Zhao et al. (2024a).

(a) Part I (Source, 6–4, 9–4, 12–1, 12–2, 12–3).

| Method | $K$ | Source | 6–4 | 9–4 | 12–1 | 12–2 | 12–3 |
|---|---|---|---|---|---|---|---|
| Baseline | 1 | 97.46% | 84.05% | 85.25% | 88.92% | 66.14% | 57.14% |
| | 3 | 97.46% | 92.43% | 92.41% | 94.12% | 67.22% | 61.43% |
| | 5 | 97.46% | 93.65% | 93.33% | 95.93% | 72.59% | 63.33% |
| PRLS-RFF | 1 | 98.17% | 91.38% | 91.23% | 90.75% | 88.17% | 90.28% |
| | 3 | 98.83% | 93.86% | 95.32% | 94.18% | 97.43% | 94.37% |
| | 5 | 98.99% | 94.54% | 96.77% | 95.63% | 98.55% | 96.13% |

(b) Part II (12–4, 15–1, 15–2, 15–3, 15–4).

| Method | $K$ | 12–4 | 15–1 | 15–2 | 15–3 | 15–4 |
|---|---|---|---|---|---|---|
| Baseline | 1 | 47.16% | 61.73% | 60.37% | 41.73% | 70.87% |
| | 3 | 58.13% | 62.14% | 66.11% | 53.67% | 71.68% |
| | 5 | 59.68% | 69.26% | 69.52% | 55.93% | 73.33% |
| PRLS-RFF | 1 | 61.43% | 79.43% | 82.47% | 78.53% | 78.33% |
| | 3 | 66.89% | 83.85% | 87.63% | 82.47% | 80.11% |
| | 5 | 68.27% | 85.77% | 89.19% | 84.11% | 81.63% |

Table 3: Cross-receiver and cross-week identification Top-1 accuracy (%) on the ManySig subset of WiSig under the DRIFT (Pan et al., 2025) protocol.

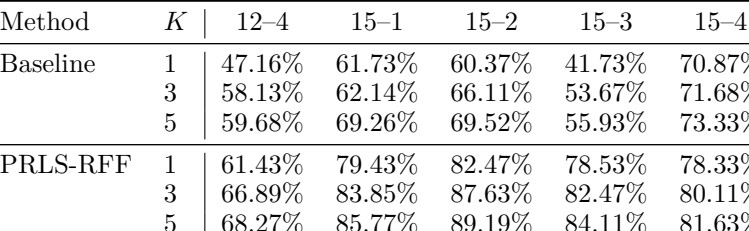

| Source Rx | Source Day | Target Rx | Target Day | DRIFT | Informer | PRLS-RFF |
|---|---|---|---|---|---|---|
| 1-1, 1-19, 8-8 | Day 1 | 14-7, 18-2, 19-2, 2-1, 2-19, 20-1, 3-19, 7-14, 7-7 | Day 2 | 80.37% | 72.41% | **87.42%** |
| | | | Day 3 | 78.05% | 81.33% | **83.57%** |
| | | | Day 4 | 79.38% | 75.41% | **80.43%** |
| | | | Average | 79.27% | 76.38% | **83.81%** |
| 1-1, 14-7, 18-2, 7-7 | Day 1 | 1-19, 18-2, 19-2, 2-1, 2-19, 20-1, 3-19, 7-14 | Day 2 | 69.70% | 81.14% | **86.41%** |
| | | | Day 3 | 71.00% | 75.31% | **90.37%** |
| | | | Day 4 | 68.30% | 64.23% | **83.42%** |
| | | | Average | 69.67% | 73.56% | **86.73%** |
| 1-1, 1-19, 14-7, 7-7, 8-8 | Day 1 | 18-2, 19-2, 2-1, 2-19, 20-1, 3-19, 7-14 | Day 2 | 79.12% | 77.21% | **89.41%** |
| | | | Day 3 | 82.10% | 81.49% | **83.19%** |
| | | | Day 4 | 81.64% | 71.14% | **87.43%** |
| | | | Average | 80.95% | 76.61% | **86.68%** |

into four time-disjoint subsets D1, D2, D3, D4. For `data#1` and `data#2`, we perform four sub-experiments by training on any three subsets and testing on the held-out subset (leave-one-subset-out across time), and report the average. We apply our 1000-epoch pretraining on the training split only, then freeze the backbone and train a linear classifier (LR $1 \times 10^{-4}$, Adam, batch size 128) for 20 epochs.

Table 1 shows that PRLS-RFF attains 98.14% on data#1 and 98.97% on data#2, surpassing the baseline. The baseline exhibits a pronounced gap between the two distance groups, with data#2 markedly higher than data#1. In contrast, our results remain uniformly high across both distance groups, demonstrating superior robustness to distance-induced domain shift.

We further adopt the more fine-grained protocol of Zhao et al. (2024a), using their D–T notation for distance–burst (e.g., 6–4 denotes 6 ft at burst 4). The source domain comprises 6–{1,2,3} and 9–{1,2,3}, split 6:2:2 into train, validation, and test, and pretraining uses only the training split. Targets include unseen time, 6–4 and 9–4, and unseen distance, 12–{1,2,3,4} and 15–{1,2,3,4}. For few-shot adaptation, we freeze the backbone and train a linear head shared across domains using $K \in \{1, 3, 5\}$ labeled samples per class, and the few-shot samples are excluded from evaluation.

Table 2 reports the results per target domain. Across all few-shot settings ($K \in \{1, 3, 5\}$), PRLS-RFF outperforms the baseline on average and matches or exceeds it on nearly all target splits. From the results, both methods perform well on the source domain and on time-only shifts (6–4, 9–4). Under larger distance shifts and compounded time with distance shifts (12–*, 15–*), the baseline degrades sharply by roughly 15–30 points. PRLS-RFF substantially narrows this gap, delivering $\geq$90% on the near-distance targets and $\geq$80% on most far-distance targets (except for 12–4) once $K \geq 3$. The 12–4 split remains challenging, yet our method still leads the baseline by 9–14 points across $K$. Across few-shot settings, performance rises markedly from 1-shot to 3-shot (about 5 points on average across targets), while gains from 3-shot to 5-shot are modest (about 1–2 points). Source-domain scores are near ceiling, so few-shot gains there are limited. These trends indicate label efficiency and robustness to distance-induced domain shift.

==On the ManySig subset of WiSig, we follow the cross-receiver and cross-day protocol of DRIFT (Pan et al., 2025). For each configuration, we select a subset of receivers as the source set (Source Rx) and use their signals from Day 1 as the training domain, while the remaining receivers (Target Rx) together with Days 2–4 form disjoint target domains, using the same setup as above. We consider several partitions, as listed in Table 3, and compare against DRIFT and Informer (Zhou et al., 2021). Across all configurations, PRLS-RFF attains the highest target accuracy, improving the average accuracy by 4.5–17.1 points over DRIFT and by 7.4–13.2 points over Informer, demonstrating strong cross-domain robustness across multiple receivers and week-scale collection gaps.==

## 4.2 Open-World Recognition

**Task definition and dataset.** The open-world setting trains on known classes only, while test inputs may include unseen classes. The system must detect unknowns (open-set rejection) and then discover their classes by clustering. Following Han et al. (2025), we adopt the same protocol and report three metrics: *Seen*, *Novel*, and *All*. *Seen* is top-1 accuracy on known classes. *Novel* is discovery accuracy on the rejected unknowns after cluster–to–class matching. *All* is overall accuracy on the union of seen and novel samples.

We evaluate on UCIHAR, the UAV RF fingerprint dataset, and the ManySig (subset of WiSig). For UCIHAR, following Han et al. (2025), we split the six classes so that three are treated as seen for training and the remaining three as novel for evaluation. For UAV RF fingerprint (Soltani et al., 2020a), we split the seven devices so that three classes are seen and the remaining four are novel. ==For WiSig (ManySig), we treat transmitters Tx6-15, Tx8-20, and Tx14-7 as seen classes and Tx14-10, Tx20-15, and Tx20-19 as novel classes.==

**Experimental setup and results.** For each dataset, we pretrain the backbone on seen classes using our method (Sec. 3), then freeze it and train a 20-epoch linear head (batch 128, LR $1 \times 10^{-5}$, Adam). Unknown detection uses the maximum softmax probability (MSP)

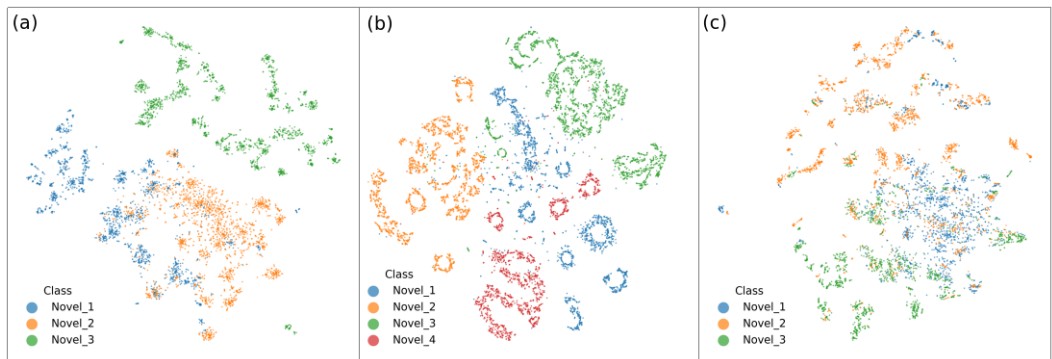

Figure 3: t-SNE of latent features $H_{\text{RFF}}$ from the frozen backbone under the open-world protocol on UCIHAR (a), UAV RF fingerprint (b), and WiSig (c).

Table 4: Open-world results on the UCIHAR dataset (3 seen / 3 novel).

| Method | Seen | Novel | All |
|---|---|---|---|
| k-means | 45.52% | 56.92% | 50.71% |
| OpenNCD | 98.00% | 84.09% | 90.52% |
| Baseline (Han et al., 2025) | 98.94% | 88.09% | 93.11% |
| **PRLS-RFF (ours)** | **99.97%** | **92.32%** | **95.43%** |

Table 5: Open-world results on the UAV RF fingerprint dataset (3 seen / 4 novel)

| Method | Seen | Novel | All |
|---|---|---|---|
| k-means | 66.01% | 32.91% | 47.35% |
| OpenNCD | 91.13% | 77.32% | 82.47% |
| Baseline (Han et al., 2025) | 92.35% | 81.42% | 84.59% |
| **PRLS-RFF (ours)** | **99.92%** | **83.47%** | **90.13%** |

Table 6: Open-world results on the ManySig (subset of WiSig) dataset (3 seen / 3 novel).

| Method | Seen | Novel | All |
|---|---|---|---|
| k-means | 43.21% | 37.21% | 40.64% |
| OpenNCD | 87.59% | 65.32% | 74.68% |
| Informer+DeepDPM | 94.21% | 43.14% | 61.28% |
| Baseline (Han et al., 2025) | 92.34% | 62.17% | 73.15% |
| **PRLS-RFF (ours)** | **97.43%** | **78.48%** | **83.47%** |

with a threshold $\tau$ (Hendrycks & Gimpel, 2017). Finally, we pass the MSP-rejected samples to DeepDPM (Ronen et al., 2022), a deep nonparametric clustering method that fits a mixture model in the learned embedding space and infers cluster assignments without fixing the number of clusters in advance.

Tables 4, 5, and 6 show that PRLS-RFF outperforms recent open-world RFF baselines on all three datasets. On UCIHAR we achieve 99.97/92.32/95.43% (Seen/Novel/All), on UAV 99.92/83.47/90.13%, and on WiSig (ManySig) 97.43/78.48/83.47%, consistently exceeding the strongest baseline in each setting. Despite near-ceiling Seen accuracy, Novel remains strong, indicating that the backbone avoids overfitting to collection-specific artifacts and yields representations that support novel-class discovery and unknown-class rejection better than existing methods.

Figure 3 visualizes $H_{\text{RFF}}$ with t-SNE for novel classes on all three datasets. UCIHAR (a) and UAV (b) show compact, well-separated clusters, while WiSig (c) still exhibits a clear clustering trend despite some overlap, reflecting its higher complexity.

### 4.3 ABLATION STUDIES

Table 7: Effect of physics-consistent augmentation views on UAV RF fingerprint dataset with a time-domain (TD) backbone.

| Variant (TD only) | Seen | Novel | All |
|---|---|---|---|
| AWGN (base view) | **96.84%** | **69.49%** | **78.12%** |
| Phase distortion (single view) | 92.37% | 65.14% | 72.42% |
| CFO (single view) | 94.52% | 61.03% | 69.54% |
| I/Q imbalance (single view) | 92.17% | 57.74% | 67.37% |
| AWGN + Phase distortion | **97.43%** | **70.57%** | **79.83%** |
| AWGN + CFO | 97.54% | 70.22% | 79.43% |
| AWGN + I/Q imbalance | 96.97% | 69.56% | 78.32% |
| **AWGN + Phase + CFO + I/Q** (full suite) | **98.96%** | **71.22%** | **80.04%** |

Table 8: Effect of the CWT on the UAV RF fingerprint dataset (TD vs. TD+CWT).

| Variant | Seen | Novel | All |
|---|---|---|---|
| TD, AWGN (base) | 96.84% | 69.49% | 78.12% |
| TD+**CWT**, AWGN (base) | 97.65% | 78.19% | 83.71% |
| TD, **Full suite** (AWGN+Phase+CFO+I/Q) | 98.96% | 71.22% | 80.04% |
| TD+**CWT**, **Full suite** | **99.92%** | **83.47%** | **90.13%** |

**Experimental setup.** We conduct ablations on the UAV RF fingerprint dataset under the open-world pipeline of Sec. 4.2. We vary two factors: (i) augmentation views on a time-domain (TD) backbone, comparing single views and their pairwise/full compositions. (ii) The backbone modality, toggling the CWT branch (TD vs. TD+CWT). Each training–evaluation run follows the same protocol: We pretrain the backbone for 1000 epochs, then freeze it and train a 20-epoch linear probe. We then use MSP for unknown detection and finally cluster the rejected samples with DeepDPM for novel discovery.

**Results.** Tables 7 and 8 report the ablations and yield two takeaways: physics-consistent multi-view augmentation improves open-set performance, and the dual-stream backbone further boosts it. For augmentation with a time-domain backbone (Table 7), we first compare single views and find that AWGN is the strongest. Building on this base, adding Phase yields the highest two-view performance among pairwise compositions. Extending to the full suite (AWGN+Phase+CFO+I/Q) produces the best overall performance. For architecture (Table 8), adding the CWT branch improves results under both the AWGN base and the full suite, confirming that time-domain and time–frequency cues are complementary for RF fingerprint representation and validating the dual-stream fusion design.

### 5 CONCLUSION

We presented PRLS-RFF, a pretraining framework for RF fingerprint representation learning. The approach fuses time and time–frequency views with a dual-stream backbone to capture complementary transient and steady-state cues, and uses physics-consistent multi-view objectives to encode physical-layer invariances rather than acquisition artifacts. Across benchmarks, the resulting representations improve cross-domain identification and open-world recognition with strong robustness. Future work will refine the backbone to improve the interpretability of the learned RF fingerprint features.

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

# A Appendix

## A.1 Backbone Architecture and Hyperparameter Ablations

In the main paper we adopt a dual-stream backbone parameterized by Mamba blocks, each implemented as a stack of four Mamba layers with $d_{\text{state}} = 256$ and model width 192. This section summarizes the empirical ablations that guided these design choices.

### A.1.1 Backbone family comparison

We compared several representative long-sequence backbones from different model architectures under the same PRLS-RFF training framework: Informer (Transformer variant for long time series), InceptionTime (CNN), GRU-FCN (RNN–CNN hybrid), and a Mamba-based backbone. All backbones were evaluated on the UAV RF fingerprint dataset in the open-world setting described in Sec. 4.2. The results are shown in Table 9.

Table 9: Open-world performance on the UAV RF fingerprint dataset with different backbone architectures, all trained under the PRLS-RFF framework.

| Backbone | Architecture | Seen (%) | Novel (%) | All (%) |
|---|---|---|---|---|
| Informer (Zhou et al., 2021) | Transformer | 95.47 | 80.37 | 83.62 |
| **PRLS-RFF (ours)** | Mamba | **99.92** | **83.47** | **90.13** |
| InceptionTime (Ismail Fawaz et al., 2020) | CNN | 78.42 | 64.13 | 71.68 |
| GRU-FCN (Shen et al., 2022) | RNN | 82.42 | 61.54 | 72.36 |

The Mamba-based backbone achieves the best accuracy and strong performance on both seen and novel classes, which empirically justifies our choice of Mamba as the default backbone family for PRLS-RFF.

Table 10: Effect of the number of layers, state dimension $d_{\text{state}}$, and model width $d_{\text{model}}$ in the Mamba backbone on the UAV RF fingerprint dataset in the open-world setting. "Params" denotes the total number of learnable parameters in the backbone.

| Layers | $d_{\text{state}}$ | $d_{\text{model}}$ | Params | Seen (%) | Novel (%) | All (%) |
|--------|--------|--------|--------|--------|--------|--------|
| 2 | 128 | 24 | 0.134496M | 76.78 | – | – |
| 2 | 128 | 48 | 0.311616M | 82.23 | – | – |
| 4 | 256 | 96 | 2.4768M | 96.14 | 79.56 | 85.37 |
| **4** | **256** | **192** | **6.336M** | **99.92** | **83.47** | **90.13** |
| 4 | 256 | 384 | 18.2016M | 98.73 | 81.16 | 87.37 |
| 6 | 256 | 384 | 27.3024M | 98.62 | 82.43 | 88.64 |
| 8 | 256 | 384 | 36.4032M | 99.43 | 82.52 | 89.62 |

### A.1.2 MAMBA BACKBONE HYPERPARAMETER ABLATION

We further ablate key architectural hyperparameters of the Mamba backbone on the same UAV open-world benchmark, varying the number of layers, the state dimension $d_{\text{state}}$, and the model width $d_{\text{model}}$, while keeping all other components of PRLS-RFF and the training setup fixed. Table 10 reports the resulting parameter counts and open-world metrics.

Moving from small 2-layer configurations with $d_{\text{state}} = 128$ and narrow $d_{\text{model}}$ to a 4-layer Mamba with $d_{\text{state}} = 256$ and $d_{\text{model}} = 192$ substantially improves open-world performance. Further increasing the model width or the number of layers markedly increases the parameter count but yields only marginal or even slightly negative changes in the accuracy on this benchmark. We therefore adopt the 4-layer $(d_{\text{state}}, d_{\text{model}}) = (256, 192)$ configuration in all main experiments as a favorable accuracy–efficiency trade-off.

Table 11: Augmentation parameter ranges (sampled i.i.d. per view unless noted). All signals are first normalized to $[-1, 1]$ component-wise; ranges are chosen to preserve baseband semantics while covering common front-end/propagation impairments.

| Operator | Symbol / Sampling | Range / Notes |
|----------|-------------------|---------------|
| AWGN | $\text{SNR}_{\text{dB}} \sim \mathcal{U}[10, 30]$ | Target SNR (dB). Splitting power equally to I/Q makes total noise power $P_n = P_x/10^{\text{SNR}/10}$. Optional: with prob. 0.2 sample from $[0, 10]$ for extra low-SNR robustness. |
| Phase distortion | $\theta[n]$ slow-varying | AR(1): $\theta[n] = \rho\,\theta[n-1] + \eta[n]$, $\rho \sim \mathcal{U}[0.95, 0.995]$, $\eta[n] \sim \mathcal{N}(0, \sigma_\eta^2)$ with $\sigma_\eta \sim \mathcal{U}[2°, 10°] \times \pi/180$ (per-step std). Caps ensure $\text{std}(\theta) \lesssim 25°$. |
| CFO | $\Delta f/f_s \sim \mathcal{U}[-5 \times 10^{-3}, 5 \times 10^{-3}]$ | Small normalized offset ($\pm 0.5\%$). To preserve symbol semantics on a length-$L$ window, also enforce $|\Delta f| \leq \phi_{\max} f_s/(2\pi L)$ with $\phi_{\max} = \pi/3$ (phase drift $\leq 60°$ over $L$). |
| I/Q imbalance | $\varepsilon_{\text{dB}} \sim \mathcal{U}[-3, 3]$, $\varphi \sim \mathcal{U}[-5°, 5°]$ | Gain/phase imbalance to build $\alpha, \beta$: $\alpha = \frac{1}{2}(1 + \varepsilon)\,e^{j\varphi/2}$, $\beta = \frac{1}{2}(1 - \varepsilon)\,e^{-j\varphi/2}$ with $\varepsilon = 10^{\varepsilon_{\text{dB}}/20}$. Controls image leakage via $|\beta/\alpha|$. |

### A.2 NON-CONTRASTIVE VS. CONTRASTIVE LOSS VARIANTS

In the main paper we use a BYOL-style non-contrastive self-supervised objective. The backbone learns by aligning physics-consistent multi-view replicas of the same RF signal with an online–target architecture and an EMA-updated target encoder (Sec. 3). In this setting, the self-supervised signal comes only from discrepancies between multiple views of the same device instance (e.g., different noise levels, phase noise, CFO, and I/Q imbalance).

Table 12: Comparison of BYOL-style non-contrastive vs. SimCLR-style contrastive pretraining on the UAV RF fingerprint dataset under the open-world protocol of Sec. 4.2.

| Loss variant | Seen | Novel | All |
|---|---|---|---|
| BYOL-style non-contrastive loss | **99.92%** | **83.47%** | **90.13%** |
| SimCLR-style contrastive loss | 99.14% | 71.56% | 82.46% |

Aligning these views encourages the model to keep view-invariant, device-intrinsic structure, hardware-level RF fingerprints, and to suppress channel- and environment-specific artifacts.

In contrast, contrastive instance-discrimination losses such as SimCLR (Tian et al., 2020) use all other samples in the batch as negatives. In realistic RF scenarios, different devices are typically observed under different channels, propagation conditions, and SNRs, so the contrastive loss can obtain large gradients by separating instances based on these non-hardware nuisance factors. This risks encoding acquisition artifacts into the backbone representation, which may be detrimental for downstream open-world self-stacking Mamba fdiscovery.

To empirically validate this choice, we implement a SimCLR-style contrastive counterpart of PRLS-RFF on the UAV dataset. We keep the backbone, physics-consistent augmentations, and pretraining hyperparameters identical to the main setup, changing only the self-supervised loss from BYOL-style non-contrastive alignment to an NT-Xent contrastive loss with in-batch negatives. After pretraining, we reuse the same open-world protocol as in Sec. 4.2 (linear head, MSP-based rejection, DeepDPM clustering) and compare Seen/Novel/All accuracies. The results in Table 12 show that both losses yield strong closed-set (Seen) performance, but the BYOL-style variant markedly improves Novel and overall accuracy, indicating that the non-contrastive formulation better supports novel-class discovery and cluster compactness than the negative-sample-based contrastive counterpart.

### A.3 Preprocessing of the UCI HAR Dataset

For the UCI Human Activity Recognition (UCI HAR) dataset, we reuse the same PRLS-RFF pipeline as for the UAV RF data and only change the input formatting. Concretely, we first reorder and concatenate the 9 sensor channels into a single real-valued sequence and treat this sequence as the in-phase (I) component. We then apply a discrete Hilbert transform to this sequence to obtain the corresponding quadrature (Q) component, which gives a pseudo-IQ complex representation. Apart from this deterministic input reformatting, the model architecture, training setup, and open-set evaluation protocol for UCI HAR are exactly the same as for the UAV RF dataset, with only the input length and number of classes being different.

### A.4 CWT Time–Frequency View: Reproducibility Details

This appendix enumerates the exact settings used to build the CWT-based time–frequency view from baseband I/Q waveforms. Waveform-level conditioning (e.g., centering, scaling) is already specified in the main text and is not repeated here.

**Dependencies and execution.** We use `numpy`, `torch` (and `torch.nn.functional` for resizing), and `fcwt` for CWT on CPU (FFTW-backed). The pipeline is deterministic given fixed inputs and package versions.

**Interface (shapes).** Input is a batch of I/Q sequences with shape $B \times T \times 2$ (the last dimension is $(I, Q)$). The output is a batch of two-channel images with shape $B \times 2 \times S \times S$, where $S = \texttt{out\_size}$ (default 256). Dtype is 32-bit float.

**Sampling rate and units.** An optional sampling rate $f_s$ is accepted: (i) if $f_s = 1$ (default), frequencies are in cycles/sample; (ii) otherwise, in Hz. We do not use the returned frequency vector in training.

**Default band and resolution (if not provided).** Given sequence length $T$ and sampling rate $f_s$:

$$f_0 = \max\left(\frac{f_s}{T},\, 10^{-6}\right), \quad f_1 = 0.48\, f_s, \quad f_n = \min(200,\, T).$$

These defaults ensure $0 < f_0 < f_1 < 0.5\, f_s$ and keep vertical resolution modest.

**Per-channel CWT.** For each example and for each channel ($I$ or $Q$), we call

$$(\mathrm{freqs}, W) \leftarrow \texttt{fcwt.cwt}(x,\, \texttt{int}(f_s),\, f_0,\, f_1,\, \texttt{int}(f_n)),$$

which returns complex coefficients $W$ of size $f_n \times T$. We use the default mother wavelet of \texttt{fcwt} (no custom parameters).

**Log-domain mapping and dynamic-range compression.** Let $M = |W|$ (magnitude). We compute an amplitude dB map:

$$S_{\mathrm{dB}} = 20 \log_{10}(M + 10^{-12}).$$

Then normalize the peak to 0,

$$S_{\mathrm{dB}} \leftarrow S_{\mathrm{dB}} - \max(S_{\mathrm{dB}}),$$

clip to a dynamic range $[-D,\, 0]$ with default $D = 80\,\mathrm{dB}$, and linearly map to $[0, 1]$:

$$I = \frac{S_{\mathrm{dB}} + D}{D} \in [0, 1]^{f_n \times T}.$$

**Resizing and stacking.** Each channel image $I$ is resized from $(f_n, T)$ to $(S, S)$ using bilinear interpolation with \texttt{align\_corners=False}. The resized $I$- and $Q$-channel images are stacked to form a two-channel tensor of shape $2 \times S \times S$ per example; batching yields $B \times 2 \times S \times S$.

**Defaults used in all experiments.** Unless noted otherwise, we use $f_s = 1$, $f_0 = f_s/T$, $f_1 = 0.48\, f_s$, $f_n = \min(200, T)$, $S = 256$, $D = 80$, bilinear resize with \texttt{align\_corners=False}, and 32-bit floats.

**Notes.** (i) The band defaults are valid for all $T \geq 2$. (ii) Peak memory per sample scales with $O(f_n T)$ for coefficients and $O(S^2)$ for the image. (iii) No stochastic components are present in this preprocessing.

## A.5 Backbone: Mamba Blocks and Fusion

This appendix records exactly how we configure and stack the Mamba blocks in our backbone.

**Blocks and layers.** Each Mamba block is implemented as a stack of four Mamba layers with identical hyperparameters (state size $d_{\mathrm{state}}$, model width $d_{\mathrm{model}}$, convolution kernel size $d_{\mathrm{conv}}$, and expansion factor $e$). The backbone uses three such blocks:

- **Time-domain block:** processes the flattened I/Q waveform sequence.
- **Time–frequency block:** processes the flattened CWT time–frequency sequence.
- **Fusion block:** takes the concatenated outputs of the two streams, and we set one Mamba block in all experiments.

**Parameterization (shared by all Mamba layers).** Let $\mathcal{M}(\cdot;\, d_{\mathrm{state}}, d_{\mathrm{conv}}, e)$ denote a Mamba layer with state size $d_{\mathrm{state}}$, depthwise-convolution kernel size $d_{\mathrm{conv}}$, and expansion factor $e$ in the internal feed-forward path. All layers share the same configuration:

$$d_{\mathrm{state}} = 256, \qquad d_{\mathrm{model}} = 192, \qquad d_{\mathrm{conv}} = 4, \qquad e = 2.$$

Residual connections are enabled (\texttt{--residual 1}). Unless stated otherwise, blocks use the default Mamba implementation settings of our codebase.

**Tokenization and streams.** Each training sample provides two 1D sequences:

- IQ stream: flattened from the baseband $I/Q$ waveform (script flag `--enc_in 2`).
- CWT stream: flattened from the two-channel CWT image.

Denote their lengths by $L_{\mathrm{IQ}}$ and $L_{\mathrm{CWT}}$ (implementation-defined by the data preprocessor). Stage A applies one $\mathcal{M}$ to each stream separately, yielding sequences of the same embedding width.

**Feature projection and fusion.** Before fusion, features are projected to a common latent width. The two per-view sequences are then concatenated along the feature axis and processed by two stacked $\mathcal{M}$ layers (Stage B).

The classifier maps the final representation to `--num_classes` categories (3 in our UAV setup).

**Other script flags.** We keep all non-architectural flags here for reproducibility:

- Number of Mambas: `--num_mambas 4`
- Projection width: `--projected_space 192`
- Padding mode: `--pad_mode repeat_head`

**Training schedule (as used in our runs).**

- Optimizer LR: `--learning_rate 1e-4`; epochs: `--train_epochs 1000`;
- Sequence lengths and batches: for `--seq_len 10240` use `--batch_size 512`;

**Reproducibility summary.** All Mamba layers are identically configured with $(d_{\mathrm{state}}, d_{\mathrm{model}}, d_{\mathrm{conv}}, e) = (256, 192, 4, 2)$. Each of the three backbone blocks (time-domain, time–frequency, and fusion) consists of four such layers with residual connections; the per-view blocks operate on the IQ and CWT flattened sequences, whose outputs are concatenated, projected to width 192, and then processed by the fusion block as in Eq. equation 10. These settings are sufficient to reinstantiate the backbone used in our experiments.