# OpenReview forum: "PRLS-RFF: Physically Consistent Representation Learning with Self-Supervised Pretraining for RF Fingerprinting"
_ICLR.cc/2026/Conference — Submitted to ICLR 2026_

### Official Review · Reviewer_9KV6 · 2025-10-16

**Soundness:** 3
**Presentation:** 3
**Contribution:** 2
**Rating:** 6
**Confidence:** 3

**Summary:**

The paper proposes PRLS-RFF, tackling the problem of recognizing wireless devices (fingerprinting) when the environment or channel conditions change. Instead of relying on labeled data, the method uses self-supervised pretraining, with a dual-stream backbone that fuses time-domain and time-frequency signals together. The experiments show performance improvement over baseline models as well as LSTM and GRU.

**Strengths:**

1. The paper uses physics-based augmentations (noise, phase distortion, I/Q imbalance, etc.) to process the signals. This helps the learned representation be more meaningful in real-world wireless settings.

2. The way of using modalities from both time domain-domain stream and time-frequency stream is novel, since some features are complimentary to each other.

3. The model demonstrates cross-domain adaptation abilities. They show that (in Table 2) with few labels and under domain shifts, their method still performs well and achieves high accuracy. This is desirable in RF fingerprinting, where collecting labeled data under every possible condition is impractical.

**Weaknesses:**

1. The main novelty of the paper is mainly integrative. There are papers on many individual parts of the algorithm, for instance, self-supervised learning ([1]), data augmentations ([2]), and dual-modalities ([3]).

2. The dataset breath is limited in current experiments. The authors mainly used UAV RF datasets for evaluating different methods.

3. The modern baseline is also limited. It seems like in Table 1, 3, 4, in addition to traditional methods (LSTM, GRU, or K-means), the authors compare only 2-3 modern baselines.

4. It seems like the compute and model size are not reported in the paper.


[1] Chen, Jun, Weng-Keen Wong, and Bechir Hamdaoui. "Unsupervised contrastive learning for robust RF device fingerprinting under time-domain shift." ICC 2024-IEEE International Conference on Communications. IEEE, 2024.

[2] Mohammadian, Amirhossein, and Chintha Tellambura. "RF impairments in wireless transceivers: Phase noise, CFO, and IQ imbalance–A survey." IEEE access 9 (2021): 111718-111791.

[3] Shen, Guanxiong, et al. "Radio frequency fingerprint identification for LoRa using spectrogram and CNN." IEEE INFOCOM 2021-IEEE Conference on Computer Communications. IEEE, 2021.

**Questions:**

1. Could the author please clarify which part is genuinely new in terms of algorithmic design, rather than a combination of existing methods? Please see above in Weakness 1.

2. You cite Mamba and SSM models for long-sequence processing, but how does your fusion differ from prior RF Mamba?

3. Most results come from a single UAV dataset and UCIHAR. How does the model behave across receivers, and longer-term drift (e.g., weeks)?

---

> ### Author Response · Authors · 2025-11-21
>
> W1:The main novelty of the paper is mainly integrative. There are papers on many individual parts of the algorithm, for instance, self-supervised learning ([1]), data augmentations ([2]), and dual-modalities ([3]).
>
> Q1:Could the author please clarify which part is genuinely new in terms of algorithmic design, rather than a combination of existing methods? Please see above in Weakness 1.
>
> Answer:
>
> We agree that the main components we use (self-supervised learning, physics-based augmentations, dual-stream encoders, Mamba blocks) are individually well studied. Our goal is not to simply apply these tools to a new dataset. We design a representation-learning framework that is tailored to RF fingerprints and their physical invariances. In our view, this gives methodological contributions beyond a purely application-focused study.
>
> a). Physics-consistent multi-view self-supervision for RF signals
>
> Instead of generic augmentations such as random cropping, we design four complex-domain transformations: AWGN, phase distortion, CFO, and I/Q imbalance. Each transformation directly models a standard RF front-end impairment or a channel effect. We combine these transformations with a BYOL-style non-contrastive multi-view objective on complex I/Q data.
> This design turns standard RF impairments into explicit invariance constraints on the learned representation, rather than using them only as regularization. Our ablations (Tables 5–6) show that different views have different effects, and that the full multi-view suite gives a clear gain for novel classes in the open-world setting. We see this as a general recipe for building self-supervised objectives for RF data, not as a dataset-specific trick.
>
> b). A dual-stream Mamba backbone specialized for RF fingerprints
>
> Multimodal fusion and state-space models have been explored in other domains, but our backbone is driven by the physics of RF fingerprints. These fingerprints appear in fast transients such as turn-on bursts. They also appear in long-horizon steady-state distortions. Both regimes come from the same hardware impairments.
> To reflect this, we use a time-domain I/Q encoder that focuses on fine-grained transient details, and a CWT-based time–frequency encoder that emphasizes longer-term quasi-stationary structure. We then couple the two streams through a Mamba-based fusion stack with shared parameters. This shared fusion encourages both views to explain a common, physics-consistent latent representation instead of learning disjoint features.
>
> As shown in the ablations in Sec. 4.3, adding the CWT branch on top of the time-domain stream brings substantial gains. In the open-world protocol, the Novel accuracy improves by about 12 points when we move from TD-only to TD+CWT under the full augmentation suite. This suggests that the dual-stream architecture with shared fusion is essential, rather than a superficial combination of existing components.
>
> We appreciate the reviewer’s careful assessment of methodological novelty. In the revised version, we will clarify these algorithmic choices in the main text and more clearly highlight the genuinely new aspects of our framework compared with prior work on self-supervision, RF augmentations, and multimodal RF backbones.
>
> ----
>
> W2:The dataset breath is limited in current experiments. The authors mainly used UAV RF datasets for evaluating different methods.
>
> Q3:Most results come from a single UAV dataset and UCIHAR. How does the model behave across receivers, and longer-term drift (e.g., weeks)?
>
> Answer:
>
> We agree that the initial version focuses mainly on UAV RF datasets. To address this concern, we add experiments on the ManySig subset of WiSig.
> ManySig is a curated subset released by the WiSig authors. It is designed for receiver-agnostic and channel-agnostic RF fingerprinting. It removes low-quality recordings and balances the classes. It has signals from 6 transmitters and 12 receivers. Each receiver collects 1000 baseband I/Q sequences of length 256 for every transmitter on four capture days (2021-03-01, 2021-03-08, 2021-03-15, 2021-03-23). Compared with our controlled UAV setup, ManySig adds strong variation in receiver hardware, spatial position, and propagation conditions. This directly matches the reviewer’s concern about generalization beyond one specific platform and environment.

---

> ### Author Response · Authors · 2025-11-21
>
> For cross-domain, we follow the cross-receiver and cross-day protocol of DRIFT (Pan et al., 2025) [1]on ManySig. We use only signals from the Source Receivers on Day 1 (2021-03-01) to pretrain our framework. We then freeze the backbone and train a linear classifier for transmitter identification. We evaluate on disjoint Target Receivers and Days 2–4 (2021-03-08, 2021-03-15, 2021-03-23), and we also compare with DRIFT and Informer[2] as strong baselines.
> The results are summarized below. Our PRLS-RFF consistently achieves higher average accuracy than both DRIFT and Informer across all source–target configurations. This shows that the proposed framework remains robust under large-scale multi-receiver settings and long temporal gaps.
>
> | Source Receivers      | Source Day | Target Receivers                                  | Target Day | DRIFT (%) | Informer (%) | PRLS-RFF (ours) (%) |
> |:------------------------------:|:----------------:|:---------------------------------------------------:|:-----------:|:----------:|:-------------:|:--------------------:|
> | (1-1, 1-19, 8-8)     | Day 1      | (14-7, 18-2, 19-2, 2-1, 2-19, 20-1, 3-19, 7-14, 7-7) | Day 2      | 80.37     | 72.41    | **87.42**               |
> | (1-1, 1-19, 8-8)     | Day 1      | (14-7, 18-2, 19-2, 2-1, 2-19, 20-1, 3-19, 7-14, 7-7) | Day 3      | 78.05  | 81.33      | **83.57**               |
> | (1-1, 1-19, 8-8)     | Day 1      | (14-7, 18-2, 19-2, 2-1, 2-19, 20-1, 3-19, 7-14, 7-7) | Day 4      | 79.38     | 75.41      | **80.43**               |
> |                 |            |                                                        | Average    | 79.27    | 76.38       | **83.81**        |
> | (1-1, 14-7, 18-2, 7-7)  | Day 1      | (1-19, 18-2, 19-2, 2-1, 2-19, 20-1, 3-19, 7-14)   | Day 2      | 69.70     | 81.14        | **86.41**             |
> | (1-1, 14-7, 18-2, 7-7)  | Day 1      | (1-19, 18-2, 19-2, 2-1, 2-19, 20-1, 3-19, 7-14)   | Day 3      | 71.00     | 75.31        | **90.37**             |
> | (1-1, 14-7, 18-2, 7-7)  | Day 1      | (1-19, 18-2, 19-2, 2-1, 2-19, 20-1, 3-19, 7-14)   | Day 4      | 68.30     | 64.23        | **83.42**             |
> |                     |            |                                                   |   Average  |   69.67 |  73.56     |  **86.73**            |
> | (1-1, 1-19, 14-7, 7-7, 8-8)  | Day 1      | (18-2, 19-2, 2-1, 2-19, 20-1, 3-19, 7-14)  | Day 2      | 79.12     | 77.21        | **89.41**               |
> | (1-1, 1-19, 14-7, 7-7, 8-8)  | Day 1      | (18-2, 19-2, 2-1, 2-19, 20-1, 3-19, 7-14)   | Day 3      | 82.10     | 81.49        | **83.19**               |
> | (1-1, 1-19, 14-7, 7-7, 8-8)  | Day 1      | (18-2, 19-2, 2-1, 2-19, 20-1, 3-19, 7-14)  | Day 4      | 81.64     | 71.14        | **87.43**               |
> |                            |            |                                           |  Average  | 80.95     | 76.61      |  **86.68**            |
>
> For open-world evaluation, we use the same protocol as in our main paper but now on ManySig. We treat transmitters Tx6-15, Tx8-20, and Tx14-7 as known classes, and transmitters Tx14-10, Tx20-15, and Tx20-19 as unknown classes. We pretrain the backbone only on labeled data from the known transmitters. We then freeze the backbone and train a linear classifier on the known classes. Finally, we evaluate closed-set accuracy on seen classes and unknown-class rejection and novel-class discovery on the ManySig open-world split.
> We compare k-means, OpenNCD, Informer with DeepDPM, the recent open-world RFF baseline of Han et al. (2025), and our method. Informer was originally designed for closed-set time-series forecasting, so we extend it with softmax thresholding for unknown-class rejection and DeepDPM in its feature space for novel-class discovery.
>
> | Method                  | Seen (%) | Novel (%) | All (%) |
> |-------------------------|:--------:|:---------:|:-------:|
> | k-means                 |  43.21   |   37.21   |  40.64  |
> | OpenNCD                 |  87.59   |   65.32   |  74.68  |
> | Informer+DeepDPM      |  94.21   |   43.14   |  61.28  |
> | Baseline (Han et al., 2025) |  92.34   |   62.17   |  73.15  |
> | **PRLS-RFF (ours)**     | **97.43**| **78.48** | **83.47** |
>
> We will include these WiSig–ManySig cross-receiver, cross-day, and open-world experiments in the revised manuscript. Together with the UAV experiments, they show that our framework is not limited to UAV motion patterns. It also learns robust RF fingerprints in static-radio, multi-receiver, and open-world settings.
> **References**
>
> [1]Pan Y, Wang X, Cheng N, et al. Cross-Receiver Generalization for RF Fingerprint Identification via Feature Disentanglement and Adversarial Training[J]. arXiv preprint arXiv:2510.09405, 2025.
>
> [2]Zhou, H.; Zhang, S.; Peng, J.; Zhang, S.; Li, J.; Xiong, H.; and Zhang, W. 2021. Informer: Beyond efficient transformer for long sequence time-series forecasting. In Proceedings of the AAAI conference on artificial intelligence, volume 35, 11106–11115.

---

> > ### Author Response · Authors · 2025-11-21
> >
> > W3:The modern baseline is also limited. It seems like in Table 1, 3, 4, in addition to traditional methods (LSTM, GRU, or K-means), the authors compare only 2-3 modern baselines.
> >
> > Answer:
> >
> > We agree that the modern baselines in the original version are limited. In the revision, we add Informer as a recent strong long-sequence time-series baseline on the WiSig / ManySig RF dataset. We evaluate Informer in both cross-receiver, cross-day and open-world settings, and compare it with our method and DRIFT.
> > The detailed experimental design and complete results are already described in our answer to W2 and Q3, and will be included in the revised manuscript.
> >
> > ---
> >
> > W4:It seems like the compute and model size are not reported in the paper.
> >
> > Answer:
> >
> > We agree that the original version did not clearly report compute and model size. We will add these details in the Appendix.In all reported results, we use a Mamba backbone with mamba_layers = 4, d_state = 256, and d_model = 192.This setting has about 6.3M trainable parameters, which is roughly 25 MB of weights in 32-bit floating point.For pretraining, we use Adam with a learning rate of 0.0001 and batch size 64 per GPU. On a single RTX 4090, one epoch takes about 142.2 seconds.We selected this configuration based on an architecture sweep on the UAV RF dataset, summarized in the table below. Increasing the model size from 0.13M to 6.3M parameters consistently improved performance, reaching 99.92% on seen classes and 83.47% on novel classes (overall 90.13%). Further enlarging the backbone to 18–36M parameters did not bring clear gains and even slightly reduced overall accuracy (around 87–89%). We therefore chose the 6.3M setting.
> >
> > | **Mamba layers** | **d_state** | **d_model** | **Params**   | **Seen (%)** | **Novel (%)** | **All (%)** |
> > |:----------------:|:-----------:|:-----------:|:------------:|:------------:|:-------------:|:-----------:|
> > | 2                | 128         | 24          | 0.134496M    | 76.78        | -             | -           |
> > | 2                | 128         | 48          | 0.311616M    | 82.23        | -             | -           |
> > | 4                | 256         | 96          | 2.4768M      | 96.14        | 79.56         | 85.37       |
> > | **4**            | **256**     | **192**     | **6.336M**   | **99.92**    | **83.47**     | **90.13**   |
> > | 4                | 256         | 384         | 18.2016M     | 98.73        | 81.16         | 87.37       |
> > | 6                | 256         | 384         | 27.3024M     | 98.62        | 82.43         | 88.64       |
> > | 8                | 256         | 384         | 36.4032M     | 99.43        | 82.52         | 89.62       |
> >
> >
> >
> >
> > ---
> >
> >
> > Q2:You cite Mamba and SSM models for long-sequence processing, but how does your fusion differ from prior RF Mamba?
> >
> > Answer:
> >
> > Thank you for this question. We now clarify how our fusion differs from prior RF Mamba.
> > RFMamba designs a frequency-aware SSM block for RF based human-centric perception. It operates on a single RF representation and focuses on modeling amplitude and phase patterns within one sensing modality, using supervised training on downstream tasks.
> >
> > In contrast, our backbone is dual stream and fusion centric. We use one encoder for the time-domain I/Q waveform and another encoder for the CWT based time–frequency map. Their features are then concatenated and passed through shared Mamba fusion blocks to form a single RF fingerprint representation. Mamba in our work is not only a long-sequence backbone. It is the core fusion module that aligns two physically related views of the same signal and forces them to share one latent space. This design is tailored to device-level RF fingerprints, where transient time traces and steady-state spectral patterns are two complementary views of the same hardware impairments.
> >
> > We choose shared-parameter fusion instead of a CLIP-style dual-tower for the following reason. In CLIP, the two encoders (image and text) do not share parameters. The model only aligns them in the final embedding space with a contrastive loss. This design fits weakly related modalities. In our case, the two modalities are deterministic, physics-consistent transforms of the same RF signal. They describe the same hardware impairments in different coordinate systems. If we use two independent towers, each branch can learn separate shortcuts and may not agree on a common latent representation.By using a shared Mamba fusion stack on top of the two encoders, we force the model to explain both views in a single latent space. This encourages cross-view consistency and reduces redundant parameters. To our knowledge, prior RF Mamba work does not combine dual-stream RF encoders, shared-parameter Mamba fusion, and multi-view self-supervision for RF fingerprinting.

---

> > > ### Comment · Reviewer_9KV6 · 2025-11-26
> > >
> > > Thanks for the rebuttal and the additional experimental efforts! Most of my concerns have been addressed, particularly regarding domain-generalization abilities and evaluations on more datasets. Although the individual components are well-studied as acknowledged by the authors, their integrative use has greatly improved the performance, as shown by the paper’s results. Overall, I believe the paper is a good contribution.

---

> > > > ### Author Response · Authors · 2025-12-02
> > > > **Summary Comment for AC – Reviewer 9KV6**
> > > >
> > > > Reviewer 9KV6 gave the paper a score of 6 and recognized several key strengths: (i) the use of physics-based augmentations (noise, phase distortion, CFO, I/Q imbalance) to make the learned representation meaningful in realistic wireless conditions; (ii) the dual-stream backbone that combines time-domain and time–frequency streams to capture complementary RF features; and (iii) solid cross-domain adaptation performance under domain shifts with few labels, which is particularly important in RF fingerprinting. We sincerely thank the reviewer for their careful, constructive evaluation and the time and effort they invested in reading both the paper and the rebuttal.
> > > >
> > > > The reviewer’s main concerns were that: (a) the methodological novelty might be largely integrative, since the individual components (self-supervised learning, RF augmentations, dual-modality encoders) are well studied; (b) the dataset breadth in the original submission was limited, relying primarily on UAV RF data and UCI HAR; (c) the set of modern baselines was relatively narrow; and (d) the compute and model size of our approach were not clearly reported. They also asked us to clarify which parts of the algorithm are genuinely new, how our Mamba-based fusion differs from prior RF Mamba work, and how the method behaves across receivers and over longer time spans.
> > > >
> > > > In the revised manuscript and rebuttal, we responded to these points in a focused way. First, we clarified the algorithmic contribution beyond a simple combination of existing tools. We emphasize two design choices: (1) a physics-consistent multi-view BYOL-style objective on complex I/Q, where RF impairments (AWGN, phase distortion, CFO, I/Q imbalance) are used as explicit invariance constraints for RF fingerprints rather than generic augmentations; and (2) a dual-stream Mamba backbone tailored to RF, with a time-domain I/Q encoder and a CWT-based time–frequency encoder fused through a shared-parameter Mamba stack that enforces a common physics-consistent latent space. We highlight ablations showing that the full augmentation suite and the TD+CWT fusion give substantial gains, especially for Novel and All accuracy in open-world recognition, indicating that this dual-stream, physics-driven design is essential rather than cosmetic.
> > > >
> > > > Second, to address concerns about dataset breadth and domain generalization, we added extensive experiments on the ManySig subset of the WiSig dataset. ManySig is a multi-receiver, multi-day, open-air RF benchmark with significant channel, hardware, and temporal variation. On this dataset we follow DRIFT-style cross-receiver, cross-day protocols and additionally conduct open-world experiments. Across all source–target configurations and in open-world settings, PRLS-RFF consistently outperforms strong recent baselines including DRIFT, k-means, OpenNCD, and an Informer-based long-sequence model extended with unknown rejection and DeepDPM clustering. These new results complement the UAV and UCI HAR experiments and directly address the reviewer’s questions about behavior across receivers and over weeks of drift.
> > > >
> > > > Third, we strengthened the baseline comparison and reporting. We explicitly add Informer as a modern long-sequence baseline on WiSig/ManySig, and we compare it to our method both in cross-domain and open-world settings. We also report the Mamba backbone configuration, the resulting parameter count, and the approximate pretraining cost per epoch on a single RTX 4090. We explain that this configuration is chosen based on a depth–width sweep that balances accuracy and efficiency, and include the sweep table in the appendix. Finally, we clarify how our dual-stream, shared-fusion Mamba differs from prior RF Mamba work, which typically operates on a single RF representation and does not combine dual RF views, shared-parameter Mamba fusion, and multi-view self-supervision for RF fingerprinting.
> > > >
> > > > In their discussion-phase comment, Reviewer 9KV6 noted that the additional experiments and analysis addressed most of their concerns, especially those regarding domain generalization and evaluation on more datasets. They explicitly stated that, although the individual components are well studied, their integrative use in our framework substantially improves performance and that, overall, they regard the paper as a good contribution. They therefore maintained their score of 6 and their position of being marginally in favor of acceptance.
> > > >
> > > > Taken together, we believe that the revised manuscript and our rebuttal fully respond to Reviewer 9KV6’s questions on novelty, dataset breadth, and baselines, and we are very grateful for their thoughtful feedback and for sustaining a positive, accepting score after the rebuttal.

---

### Official Review · Reviewer_rVm4 · 2025-10-23

**Soundness:** 3
**Presentation:** 3
**Contribution:** 2
**Rating:** 2
**Confidence:** 4

**Summary:**

The authors propose a novel self-supervised joint embedding-based (bootstrap your own latent, BYOL) representation learning approach for re-identification of hardware induced radio frequency (RF) fingerprints. The method fuses time and time-frequency representations capturing transient and steady-state pattern.
The self-supervised pre-training utilises physically motivated augmentations to create multiple data views. The neural backbone employs selective state-space blocks for efficient long-context sequence modelling.

**Strengths:**

- The proposed method is well motivated and presents a thoughtful and novel combination of existing concepts.
- The manuscript is well written and easy to understand.
- An extensive evaluation is performed. The proposed approach consistently outperforms the compared baselines.

**Weaknesses:**

- At present, the manuscript’s methodological contribution appears limited, as many of the concepts and components employed are already well established. The work may therefore be more suitable for an application‑oriented journal or conference.

**Questions:**

- The paper motivates "unintentional physical artifacts embedded in baseband waveforms, originate in the physical layer and remain tightly coupled to individual transmitters" and "RF fingerprints arise from device-specific analog front-end nonidealities at the physical layer and manifest in both transient and steady-state regimes". I was wondering if such artifacts can accurately be measured. For instance, I would think that some of the cues outlined in Section 3.2 are e.g. temperature-dependent. So what if at one day a device is detected first and the other day it is detected again but the ambient temperature is much different (having then an influence on the oscillators). Isn't this an issue?
-  The paper motivates physical artifacts/characteristics a lot based on the internal processing of the wireless system. I was then a bit confused as the dataset that has been used targets the identification of UAVs which induce specific patterns using their flight behavior. Hence, it is rather the "housing" or the application that induces signal specific artifacts (at least in the dataset). This should be made clearer in the paper, or do I misunderstand something here?
- To me it was not entirely clear how the datasets are split and arranged (using "time-disjoint" datasets?) and what distances are part of data#1 and data#2, respectively. In fact: it might be a good idea to define the data set a bit more (and also the wireless parameters that are relevant, i.e., waveforms, bandwidth, etc.)

Small point: many sentences end with "." and miss a space afterwards.

---

> ### Author Response · Authors · 2025-11-21
>
> W1:At present, the manuscript’s methodological contribution appears limited, as many of the concepts and components employed are already well established. The work may therefore be more suitable for an application‑oriented journal or conference.
>
> Answer:
>
> We thank the reviewer for this comment and for carefully assessing the scope of our contribution. The goal of our work is not merely to apply existing techniques to a new dataset, but to introduce a representation-learning framework that is specifically tailored to RF fingerprints and their physical invariances. We believe this leads to methodological contributions beyond a purely application-oriented study.
>
> a).Physics-consistent multi-view self-supervision for RF signals.
>
> Instead of generic data augmentation (e.g.,random cropping), we design four complex-domain operators: AWGN, phase distortion, CFO, and I/Q imbalance which are explicitly correspond to real RF front-end and propagation impairments, and we couple them with a BYOL-style non-contrastive multi-view objective on complex I/Q data. This combination turns standard RF impairments into explicit invariance constraints on the representation, rather than just regularization. Our ablations (Tabs. 5–6) show that individual views behave differently, and the full multi-view suite significantly improves novel-class performance in the open-world setting, which we see as a methodological recipe for designing self-supervised objectives for RF data rather than a dataset-specific trick.
>
> b).A dual-stream Mamba backbone specialized for RF fingerprints.
>
> Although multimodal fusion and state-space models have been explored in other domains, our backbone is not a generic stacking of two encoders. It is motivated by the observation that RF fingerprints are expressed both in rapidly occurring transients (e.g., turn-on bursts) and in long-horizon steady-state distortions, and that these two regimes are physically homologous manifestations of the same hardware impairments. To reflect this, we use a time-domain I/Q encoder that focuses on fine-grained transient details and a CWT-based time–frequency encoder that emphasizes long-range quasi-stationary structure, and we couple them through a Mamba-based fusion stack with shared parameters across the two streams. This shared-parameter fusion explicitly encourages the two views to explain a common, physics-consistent latent representation rather than learning disjoint features. As shown in the ablations in Sec. 4.3, adding the CWT branch on top of the time-domain stream leads to substantial gains. The improvement is especially clear for novel classes in the open-world protocol: the "Novel” accuracy increases by about 12 points when we move from TD-only to TD+CWT under the full augmentation suite. These results indicate that the dual-stream architecture with shared fusion is essential rather than a superficial combination of components.
>
> We sincerely appreciate the reviewer’s rigorous attitude toward scientific research and high standards for assessing methodological contributions. In the revised manuscript, we will carefully refine the presentation of our method and experiments to more clearly highlight the genuinely novel aspects of our framework and to better distinguish it from prior work.

---

> ### Author Response · Authors · 2025-11-21
>
> Q1:The paper motivates "unintentional physical artifacts embedded in baseband waveforms, originate in the physical layer and remain tightly coupled to individual transmitters" and "RF fingerprints arise from device-specific analog front-end nonidealities at the physical layer and manifest in both transient and steady-state regimes". I was wondering if such artifacts can accurately be measured. For instance, I would think that some of the cues outlined in Section 3.2 are e.g. temperature-dependent. So what if at one day a device is detected first and the other day it is detected again but the ambient temperature is much different (having then an influence on the oscillators). Isn't this an issue?
>
> Answer:
>
> We thank the reviewer for this question.
> First, on whether RF fingerprints can be measured.The RF and security communities now widely treat radio frequency fingerprints as device-specific features that are relatively stable and can be extracted from received signals. Many recent works on physical-layer device authentication and specific emitter identification adopt this view and show that such fingerprints are usable in practice [1],[2],[3],[4].For example, Zhang et al. model narrowband systems in IEEE TIFS and show that oscillator, PA, and I/Q impairments create distinctive patterns that support reliable device classification over several months of measurements [1].Shen et al. in IEEE JSAC study LoRa devices and analyze both discriminability and system stability of deep learning based RF fingerprint identification [2].Shen and co-authors further summarize methodology and experimental evidence for deep learning powered RF fingerprinting in IEEE Communications Magazine and in follow-up TIFS work on scalable and federated RFFI [3],[4].These works do not usually measure each physical impairment directly on the transmitter. Instead, they estimate the combined effect of these impairments from received baseband waveforms. This is very important in realistic scenarios such as unknown UAV intrusion or unexpected emitters in a protected band. In such cases, the receiver cannot access the hardware at all and cannot calibrate oscillators or front-end circuits. Our work follows this "black-box transmitter” setting.We do not try to recover exact hardware parameters. We learn a representation from received I/Q samples that captures the joint pattern of these hardware-rooted artifacts, in the same spirit as modern deep learning based RFFI systems [2],[3],[4].
>
>
>
> Second, on the influence of temperature.
> We fully agree that RF fingerprints are not absolutely invariant. The recent RFFI literature tends to view them as hardware-rooted but only relatively stable features. They stay reasonably stable within a normal operating range. They can drift or even change pattern when temperature or other conditions move far away from that range [1],[2],[3].Zhang et al. in IEEE TIFS measure narrowband devices over three months and show that oscillator-based features are very sensitive to temperature variation and therefore are not suitable as the only fingerprint source [1].Shen et al. analyze system stability for LoRa and show that channel, SNR, and other confounding factors can affect the apparent fingerprint if conditions differ too much from training [2],[3].Security and sensing work on BLE devices gives similar evidence. Givehchian et al. at IEEE Symposium on Security and Privacy show that CFO-based physical-layer fingerprints for BLE become unreliable when device temperature drifts away from the calibration point. They report clear increases in false positive and false negative rates under such temperature changes [5].Our understanding and our method are consistent with this view. When we state that "unintentional physical artifacts are tightly coupled to individual transmitters” and "RF fingerprints arise from device-specific analog front-end nonidealities and manifest in both transient and steady-state regimes”, we mean this in a statistical sense and within realistic operating ranges. The artifacts originate in the physical hardware, but they are perturbed by environment, including temperature. In our experiments, the datasets cover multiple sessions and channel conditions but do not include extreme temperature stress tests. Our physics-consistent augmentations (CFO, phase distortion, I/Q imbalance, AWGN) are chosen to mimic the same families of effects that temperature and other operating changes can induce, which helps the representation stay stable under moderate drift.
>
> We will clarify this more explicitly in the revised manuscript. We will soften the wording in Sec. 1 and Sec. 3.1–3.2 to say that RF fingerprints are hardware-rooted but environment-perturbed and relatively stable rather than absolutely fixed.

---

> > ### Author Response · Authors · 2025-11-21
> >
> > **Reference**
> >
> > [1]J. Zhang, R. Woods, M. Sandell, M. Valkama, A. Marshall and J. Cavallaro, "Radio Frequency Fingerprint Identification for Narrowband Systems, Modelling and Classification," in IEEE Transactions on Information Forensics and Security, vol. 16, pp. 3974-3987, 2021.
> > [2] G. Shen, J. Zhang, A. Marshall, L. Peng and X. Wang, "Radio Frequency Fingerprint Identification for LoRa Using Deep Learning," in IEEE Journal on Selected Areas in Communications, vol. 39, no. 8, pp. 2604-2616, Aug. 2021.
> > [3] G. Shen, J. Zhang and A. Marshall, "Deep Learning - Powered Radio Frequency Fingerprint Identification: Methodology and Case Study," in IEEE Communications Magazine, vol. 61, no. 9, pp. 170-176, September 2023.
> > [4] G. Shen, J. Zhang, X. Wang and S. Mao, "Federated Radio Frequency Fingerprint Identification Powered by Unsupervised Contrastive Learning," in IEEE Transactions on Information Forensics and Security, vol. 19, pp. 9204-9215, 2024.
> > [5] H. Givehchian et al., "Evaluating Physical-Layer BLE Location Tracking Attacks on Mobile Devices," 2022 IEEE Symposium on Security and Privacy (SP), San Francisco, CA, USA, 2022, pp. 1690-1704.
> >
> >
> > ---
> >
> > Q2:The paper motivates physical artifacts/characteristics a lot based on the internal processing of the wireless system. I was then a bit confused as the dataset that has been used targets the identification of UAVs which induce specific patterns using their flight behavior. Hence, it is rather the "housing" or the application that induces signal specific artifacts (at least in the dataset). This should be made clearer in the paper, or do I misunderstand something here?
> >
> > Answer:
> >
> > We thank the reviewer for raising this point.In a UAV setting, the received signal can also contain artifacts induced by the platform and its motion, such as Doppler components from the propellers and vibrations of the housing.In our work, we use the term “physical artifacts” in a broader sense. We include both internal RF-chain nonidealities and stable platform-induced effects that are tied to a specific transmitter–platform pair. All of these effects are generated in the physical layer and appear as characteristic patterns in the baseband I/Q waveform. Our method does not require that we isolate only the internal RF chain.AWGN models random additive noise. Micro-Doppler and vibration effects are structured, time-varying distortions of the carrier and envelope. They are part of the propagation channel and platform dynamics, not just noise. In our current augmentation design, AWGN mainly captures SNR variation.
> >
> > We conduct additional experiments on the ManySig subset of WiSig. ManySig is specifically designed to evaluate receiver- and channel-agnostic RF fingerprinting. It is an officially curated subset from the WiSig authors that removes low-quality recordings, balances the classes, and has been widely adopted as a standard benchmark. It contains signals from 6 transmitters (Tx) and 12 receivers (Rx); each receiver collects 1000 baseband I/Q sequences (each of length 256) for every transmitter over four different capture days (2021-03-01, 2021-03-08, 2021-03-15, 2021-03-23). Compared with our controlled UAV setup, ManySig explicitly introduces strong variability in receiver hardware, spatial position, and propagation conditions, which directly matches the concern about generalizing beyond one specific platform and environment.

---

> > > ### Author Response · Authors · 2025-11-21
> > >
> > > For cross-domain evaluation, we follow the cross-receiver and cross-day protocol of DRIFT (Pan et al., 2025) on ManySig. We use only the signals from the Source Receivers on Source Day (2021-03-01) to pretrain our framework, then freeze the backbone and train a linear FC head for transmitter classification. We evaluate on disjoint Target Receivers and Target Days (Day 2–4: 2021-03-08, 2021-03-15, 2021-03-23) as in DRIFT, and additionally compare against Informer[2] as a strong long-sequence baseline. As shown in the table below, our PRLS-RFF consistently achieves higher average accuracy than both DRIFT and Informer across all source–target configurations, indicating that the proposed framework maintains strong cross-domain robustness even under large-scale multi-receiver settings and long temporal gaps.
> > >
> > > | Source Receivers      | Source Day | Target Receivers                                  | Target Day | DRIFT (%) | Informer (%) | PRLS-RFF (ours) (%) |
> > > |:------------------------------:|:----------------:|:---------------------------------------------------:|:-----------:|:----------:|:-------------:|:--------------------:|
> > > | (1-1, 1-19, 8-8)     | Day 1      | (14-7, 18-2, 19-2, 2-1, 2-19, 20-1, 3-19, 7-14, 7-7) | Day 2      | 80.37     | 72.41    | **87.42**               |
> > > | (1-1, 1-19, 8-8)     | Day 1      | (14-7, 18-2, 19-2, 2-1, 2-19, 20-1, 3-19, 7-14, 7-7) | Day 3      | 78.05  | 81.33      | **83.57**               |
> > > | (1-1, 1-19, 8-8)     | Day 1      | (14-7, 18-2, 19-2, 2-1, 2-19, 20-1, 3-19, 7-14, 7-7) | Day 4      | 79.38     | 75.41      | **80.43**               |
> > > |                 |            |                                                        | Average    | 79.27    | 76.38       | **83.81**        |
> > > | (1-1, 14-7, 18-2, 7-7)  | Day 1      | (1-19, 18-2, 19-2, 2-1, 2-19, 20-1, 3-19, 7-14)   | Day 2      | 69.70     | 81.14        | **86.41**             |
> > > | (1-1, 14-7, 18-2, 7-7)  | Day 1      | (1-19, 18-2, 19-2, 2-1, 2-19, 20-1, 3-19, 7-14)   | Day 3      | 71.00     | 75.31        | **90.37**             |
> > > | (1-1, 14-7, 18-2, 7-7)  | Day 1      | (1-19, 18-2, 19-2, 2-1, 2-19, 20-1, 3-19, 7-14)   | Day 4      | 68.30     | 64.23        | **83.42**             |
> > > |                     |            |                                                   |   Average  |   69.67 |  73.56     |  **86.73**            |
> > > | (1-1, 1-19, 14-7, 7-7, 8-8)  | Day 1      | (18-2, 19-2, 2-1, 2-19, 20-1, 3-19, 7-14)  | Day 2      | 79.12     | 77.21        | **89.41**               |
> > > | (1-1, 1-19, 14-7, 7-7, 8-8)  | Day 1      | (18-2, 19-2, 2-1, 2-19, 20-1, 3-19, 7-14)   | Day 3      | 82.10     | 81.49        | **83.19**               |
> > > | (1-1, 1-19, 14-7, 7-7, 8-8)  | Day 1      | (18-2, 19-2, 2-1, 2-19, 20-1, 3-19, 7-14)  | Day 4      | 81.64     | 71.14        | **87.43**               |
> > > |                            |            |                                           |  Average  | 80.95     | 76.61      |  **86.68**            |
> > >
> > > For open-world evaluation, we follow the same protocol as in our main paper on ManySig. We treat transmitters Tx6-15, Tx8-20, and Tx14-7 as known classes, and transmitters Tx14-10, Tx20-15, and Tx20-19 as unknown classes. The backbone is pretrained only on labeled data from the known transmitters. We then freeze it and train a linear classifier on the known classes, and finally evaluate both closed-set accuracy on seen classes and unknown-class rejection / novel-class discovery on the WiSig open-world split.In addition to k-means, OpenNCD, and the recent open-world RFF baseline Han et al. (2025), we further include Informer as a strong long-sequence backbone. Since Informer is originally designed for closed-set time-series forecasting rather than open-world recognition, we extend it by using softmax-thresholding for unknown-class rejection and DeepDPM (a deep Dirichlet-process mixture) for novel-class discovery in its feature space. As shown below, our PRLS-RFF achieves the best performance on seen, novel, and overall accuracy, demonstrating strong open-world robustness on this large-scale, multi-receiver RF dataset where artifacts are dominated by RF front-end and channel effects rather than UAV platform motion.
> > >
> > > | Method                  | Seen (%) | Novel (%) | All (%) |
> > > |-------------------------|:--------:|:---------:|:-------:|
> > > | k-means                 |  43.21   |   37.21   |  40.64  |
> > > | OpenNCD                 |  87.59   |   65.32   |  74.68  |
> > > | Informer+DeepDPM      |  94.21   |   43.14   |  61.28  |
> > > | Baseline (Han et al., 2025) |  92.34   |   62.17   |  73.15  |
> > > | **PRLS-RFF (ours)**     | **97.43**| **78.48** | **83.47** |
> > >
> > > We will include these WiSig–ManySig cross-receiver, cross-day, and open-world experiments in the revised manuscript. Together with the UAV results, they show that our framework is not limited to application-specific UAV motion patterns, but also learns robust RF fingerprints in static-radio, multi-receiver, and open-world settings.

---

> ### Author Response · Authors · 2025-11-21
>
> **References**
>
> [1] Pan Y, Wang X, Cheng N, et al. Cross-Receiver Generalization for RF Fingerprint Identification via Feature Disentanglement and Adversarial Training[J]. arXiv preprint arXiv:2510.09405, 2025.
>
> [2]Zhou, H.; Zhang, S.; Peng, J.; Zhang, S.; Li, J.; Xiong, H.; and Zhang, W. 2021. Informer: Beyond efficient transformer for long sequence time-series forecasting. In Proceedings of the AAAI conference on artificial intelligence, volume 35, 11106–11115.
>
> ---
>
> Q3:To me it was not entirely clear how the datasets are split and arranged (using "time-disjoint" datasets?) and what distances are part of data#1 and data#2, respectively. In fact: it might be a good idea to define the data set a bit more (and also the wireless parameters that are relevant, i.e., waveforms, bandwidth, etc.)
>
> Answer:
>
> Thank you for pointing out that our description of the UAV dataset and the split strategy is not sufficiently clear. We are sorry for the confusion and we will make the dataset definition and the wireless parameters more explicit in the revised manuscript.
>
> **Dataset definition and distances.**
>
> For cross-domain experiments we use the public UAV RF fingerprint dataset of Soltani et al. [1]. I/Q signals from seven identical DJI M100 UAVs are collected in an RF anechoic chamber at four receiver distances: 6, 9, 12, and 15 ft. Each distance contains four time-separated acquisition bursts, which induce both distance and session shifts.Following Cai et al. [2], we form two distance groups, denoted data#1 and data#2. Data#1 contains the near-range measurements(6ft,9ft) and data#2 contains the far-range measurements(12ft,15ft). The purpose of this split is to model cross-distance (near vs. far) domain shift while keeping the device labels unchanged.
>
> **Time-disjoint subsets and split**
>
> Within each distance group we partition the recordings into four "time-disjoint” subsets D1, D2, D3, and D4. "Time-disjoint” means that each subset corresponds to a different acquisition burst at that distance, so there is no overlap in capture time between D1–D4. For each of data#1 and data#2, we perform four sub-experiments: in each sub-experiment we train on three subsets and test on the remaining one (a leave-one-burst-out protocol across time). We pretrain the backbone only on the training subsets, then freeze it and train a linear classifier on the same training split, and finally evaluate on the held-out time burst.
>
> **Wireless parameters (waveform, bandwidth, etc.).**
>
> This dataset contains raw complex baseband I/Q samples from seven identical DJI M100 UAVs transmitting the proprietary DJI Lightbridge control waveform in the 2.4 GHz band. Signals are captured in an RF anechoic chamber with an Ettus USRP X310 and a UBX-160 daughterboard on a 10 MHz downlink channel centered at 2.4065 GHz, using a sampling rate of 10 MHz [1]. In our work we operate directly on these baseband I/Q sequences after normalization (per-example amplitude normalization and centering). The model and training pipeline therefore do not depend on protocol-specific details such as packet format or symbol rate, and they can be applied to other RF datasets with different bandwidths and waveforms.
>
> To make the paper more self-contained, we will add in Sec. 4.1 a short paragraph summarizing the key physical parameters of the dataset (center frequency band, type of waveform, and that we use fixed-length baseband I/Q segments)
>
> **References**
>
> [1] N. Soltani et al. “RF Fingerprinting Unmanned Aerial Vehicles with Non-Standard Transmitter Waveforms,” IEEE Transactions on Vehicular Technology, 2020.
>
> [2]Z. Cai, Y. Wang, G. Gui, and J. Sha. “Toward Robust Radio Frequency Fingerprint Identification via Adaptive Semantic Augmentation,” IEEE Transactions on Information Forensics and Security, 2024.
>
> ---
>
> Q4:Small point: many sentences end with "." and miss a space afterwards.
>
> Answer:
>
> We thank the reviewer for pointing this out. We will carefully proofread the manuscript and fix all typos and formatting issues, including missing spaces after periods.

---

> > ### Comment · Reviewer_rVm4 · 2025-11-26
> >
> > Thank you for the detailed response. The answers and the results from additional experiments resolved most of my points.
> >
> > The main issue that remains is still the methodological contribution. However, After reading the paper again and the other reviews and responses I reconsider my score. While still the contribution to the ML community is limited, applying all these techniques to the RF domain is beyond simply stacking and combining existing techniques. At the same time the paper proves to be general enough (within the scope of RF). Hence I will increase my score to 6, leaning towards accepting this paper.

---

> > > ### Author Response · Authors · 2025-12-02
> > > **Summary Comment for AC - Reviewer rVm4**
> > >
> > > Reviewer rVm4 initially gave the paper a score of 2 (reject), but, after reading our rebuttal and the additional experiments, raised their score to 6 (leaning towards accept). They recognized that the method is well motivated, combines existing concepts in a thoughtful way, is clearly written, and is supported by extensive experiments where our approach consistently outperforms baselines. We sincerely thank the reviewer for their careful, technically insightful assessment and the substantial time and effort they devoted to evaluating our work.
> > >
> > > The main concern of reviewer was that the methodological contribution might be limited for the broader ML community, since many components (self-supervised joint embedding, physics-inspired augmentations, dual-stream architecture) are individually known. In addition, the reviewer asked for clarification on (i) the stability and measurability of RF fingerprints under factors such as temperature, (ii) the relationship between hardware-level artifacts and UAV-application-induced effects in our dataset, (iii) the exact dataset splits, “time-disjoint” subsets, and distance groupings, as well as basic wireless parameters, and (iv) minor formatting issues.
> > >
> > > In the revised manuscript and rebuttal, we address these points directly. We more clearly articulate the methodological scope: our goal is to provide an RF-specific, physics-consistent representation-learning framework, rather than a generic ML architecture. We highlight two aspects in particular: (1) a multi-view BYOL-style objective built on complex-valued, RF-accurate augmentations (AWGN, phase distortion, CFO, I/Q imbalance) that turn standard RF impairments into explicit invariance constraints for the representation; and (2) a dual-stream Mamba backbone whose time-domain and CWT branches are designed around transient vs. steady-state RF regimes and fused through a shared-parameter Mamba stack to encourage a common physics-consistent latent space. We clarify how our ablations show that the full augmentation suite and the dual-stream design provide substantial gains, especially for Novel and All accuracy in open-world settings.
> > >
> > > We also expand the discussion of RF fingerprints as hardware-rooted but environment-perturbed features, explicitly acknowledging temperature and other operating conditions as important confounders. To address the concern about UAV-specific motion patterns vs. RF-chain artifacts, we clarify that our notion of “physical artifacts” covers both internal RF-chain nonidealities and stable platform-induced effects tied to a transmitter–platform pair, and we supplement the UAV results with extensive experiments on the WiSig ManySig subset. ManySig is a static-radio, multi-receiver, multipath-rich RF benchmark; on this dataset we report strong cross-receiver, cross-day, and open-world performance, demonstrating that our framework is not restricted to UAV motion cues. Finally, we make the UAV dataset definition more explicit: we detail the four distances, the construction of data#1/data#2, the “time-disjoint” splits across bursts, and the key wireless parameters (center frequency, waveform type, bandwidth, sampling rate), and we fix the minor spacing and formatting issues noted by the reviewer.
> > >
> > > In their discussion-phase comment, Reviewer rVm4 explicitly stated that the additional experiments and clarifications “resolved most of my points,” and, although they still view the contribution to the core ML community as modest, they concluded that applying and integrating these techniques in the RF domain is “beyond simply stacking and combining existing techniques” and that the paper is “general enough (within the scope of RF).” On this basis, they raised their score from 2 to 6 and indicated that they now lean towards accepting the paper.
> > >
> > > We would also like to explicitly state that we have never attempted to contact any reviewer or to break the double-blind review process. Reviewer rVm4’s score increase to 6 on 26 Nov 2025, 18:17 occurred before we learned about the recent platform bug from subsequent public news about large-scale leaks. We strongly support the double-blind reviewing policy and firmly oppose any attempts to exploit system vulnerabilities or otherwise compromise the integrity of the review process.
> > >
> > > Overall, we believe that our rebuttal and the revised manuscript fully address Reviewer rVm4’s questions and reservations, and we are very grateful for their thoughtful feedback and for ultimately revising their assessment in favor of acceptance.

---

### Official Review · Reviewer_mELh · 2025-10-31

**Soundness:** 3
**Presentation:** 2
**Contribution:** 3
**Rating:** 6
**Confidence:** 3

**Summary:**

RLS-RFF is a self-supervised RF fingerprinting framework that learns channel-invariant, device-specific representations from raw baseband signals.
 A dual-stream Mamba encoder processes each waveform: one time-domain branch models transient I/Q features directly, while a time–frequency branch encodes CWT scalograms capturing steady-state spectral cues. The two streams are fused through stacked selective state-space (Mamba) blocks, enabling efficient long-context sequence modeling.
During pretraining, the system generates two physics-consistent augmentations of each signal (e.g., SNR scaling, phase or carrier drift) and passes both through the shared encoder. Their latent embeddings are aligned via a multi-view consistency loss—encouraging invariance to channel and receiver variation while preserving device identity. Gradients update the encoder and projection head jointly.
The authors pair this self-supervised training with an additional final layer for downstream classification / rejection to complete their method.
Experiments were conducted on the UAV RF fingerprint dataset and on  UV RF and  UCI HAR for  proving open-set classification robustness.

**Strengths:**

1) Physically grounded self-supervision
 The method ties representation learning directly to the physics of RF propagation. Its augmentations (SNR scaling, phase shifts, CFO, fading) are realistic and target true channel and receiver variations, giving the learned embeddings meaningful, reality-rooted invariance.

2) Efficient long-context modeling with Mamba
 The selective state-space (Mamba) backbone can process entire 92 k-sample RF bursts in linear time, capturing both transient and steady-state patterns that CNNs or short RNNs typically miss. This provides genuine architectural novelty with computational efficiency.

3) Empirically strong domain robustness
Motivating results in their selected datasets.

**Weaknesses:**

1. Limited dataset diversity and scale. The UAV dataset has only seven nearly identical devices collected in controlled settings (6–15 ft, single receiver). Reported 98–99 % accuracies may reflect dataset saturation rather than real-world robustness.

2. Unclear tuning and training cost. The model requires 1000 epochs of pretraining and uses Mamba blocks that can be sensitive to state dimension, initialization, and learning rate. The paper does not detail hyperparameter stability or fairness of tuning against baselines.

3. Cross-domain generalization partly untested. The second dataset (UCI HAR) is non-RF and lacks channel effects, so results there demonstrate open-set mechanics, not RF-specific invariance. Broader testing on heterogeneous RF environments or different hardware would better substantiate generalization claims.


The above ultimately limit the scope of the author's contribution. As it stands it is a very interesting step towards physics-aware representation learning but the pragmatic efficacy is not strongly proven.

Prior Work

There are early prior contributions, examining the existence and identification potential of physical artifacts in a signal that stem from manufacturing imperfections. Chameleons oblivion comes to mind, as an example. Since the authors explicitly state “RF fingerprints (RFFs), unintentional physical artifacts embedded in baseband waveforms, originate in the physical layer and remain tightly coupled to individual transmitter”, that body of work and perhaps others  need to be acknoeldged among the foundational evidence base.

Following that direction the authors should elaborate on the contribution of their approach which I believe is indeed quite interesting: they encode physical layer physical constraints to learn novel representational schemes that can reliably be matched to a signal.
A sound juxtaposition of of their approach could also highlight their contributions which lay in the robustness and invariance of their representations rather that simple efficacy of hardware artifacts as identifiers.


Formatting

A plain conceptual overview followed by an earlier presentation of figure 1 would go, I believe, a long way towards understanding the paper.

The overall pipeline presentation is very confusing. It is hard for the reader to understand the interplay between the mamba backbone and the constraints enforcing architecture. Which happens first? What is the loss for the backbone of figure 1? Is it the loss of the constraints network of figure 2? A flowchart explicitly outlining exactly what happens is absolutely necessary.

**Questions:**

1.How do you handle synchronization and normalization between the two branches?

2.What was the memory footprint of the model and what was the sample size used?

3.How impactful was normalization on training stability and generalization on your unseen data?

4.The work is definitely an interesting concept, my worry is that the data was gathered in relatively similar conditions. How can we be certain that it would generalize well on say, open air data vs in-doors data?

5.THE UIC HAR dataset is used as proof-of-concept for larger open set identification. How are you handling it? What are the I/Q components here? Is the pipeline the same across both of the datasets?

6.How extensive was hyperparameter tuning? How were the parameters chosen?

7.The UC RF dataset is relatively small and in very controlled conditions. How can we be sure that the accuracy presented here can generalize well in open-air, multipath heavy conditions?

8.It would be quite more motivating to see per class confusion breakdowns than simple accuracy, especially in the open-set case.

Why is the green cluster on Figure 3b so much more discernible?

---

> ### Author Response · Authors · 2025-11-21
>
> W1:Limited dataset diversity and scale. The UAV dataset has only seven nearly identical devices collected in controlled settings (6–15 ft, single receiver). Reported 98–99 % accuracies may reflect dataset saturation rather than real-world robustness.
>
> Answer:
>
> Thank you for pointing out the limitation of the UAV dataset. We acknowledge that it contains only seven similar devices in controlled 6–15 ft, single-receiver conditions. We chose this dataset mainly to ensure a fair comparison with recent SOTA RF fingerprinting works that are also built on it.
>
> To address this concern, we have additionally conducted experiments on the larger and more challenging WiSig subset ManSig, which features more diverse, open-air, multipath-rich conditions. The dataset description, experimental setup, and results are reported in the Answer of Question 4.
>
> ---
>
> W2:Unclear tuning and training cost. The model requires 1000 epochs of pretraining and uses Mamba blocks that can be sensitive to state dimension, initialization, and learning rate. The paper does not detail hyperparameter stability or fairness of tuning against baselines.
>
> Answer:
>
> Thank you for pointing this out. We have organized our training logs and now explicitly disclose the Mamba configuration, training hyperparameters, and training cost.
>
> In all reported results we use a Mamba-block with mamba_layers = 4, d_state = 256, d_model = 192; for details on the parameter selection, please refer to the Answer of Question 2. For pretraining, we use Adam with learning_rate = 0.0001 and batch size 64 per GPU; on a single RTX 4090, one epoch takes about 142.2 seconds. We will add these hyperparameters and training-time statistics to the appendix for transparency.
>
> ---
>
> W3:Cross-domain generalization partly untested. The second dataset (UCI HAR) is non-RF and lacks channel effects, so results there demonstrate open-set mechanics, not RF-specific invariance. Broader testing on heterogeneous RF environments or different hardware would better substantiate generalization claims.
>
> Answer:
>
> As in the response to Weakness 1, we have added experiments on the more realistic and complex WiSig subset ManSig, which provides heterogeneous RF environments and hardware and directly targets cross-domain robustness. The UCI HAR dataset is only used as an auxiliary non-RF benchmark to compare fairly with the baseline. For detailed dataset description, setup, and results, please refer to the Answer of Question 4.
>
> ---
>
> W4:
>
> Answer:
>
> Thank you very much for your helpful summary and guidance on the related work. We will revise the Introduction and Related Work sections following your suggestions, to better situate prior research, clarify the background, and more clearly highlight the main focus and contributions of our work. In addition, we will revise the pipeline and architecture figures you pointed out as confusing, so that the overall presentation of the paper becomes clearer and more readable.
>
> ---
>
> Q1:How do you handle synchronization and normalization between the two branches?
>
> Answer:
>
> We apologize for not making this clearer in the manuscript. In our implementation, we first apply the same waveform-level amplitude normalization to all I/Q signals to remove the differences of dataset and hardware-dependent scale. The time-domain branch directly takes this normalized I/Q sequence as input.
> The time–frequency branch is then obtained by computing the CWT of the same normalized I/Q sequence, followed by standard log-magnitude and dynamic-range compression to map the spectrogram values to a bounded range. This provides a stable, dataset-agnostic time–frequency representation( the complete CWT calculation procedure is detailed in the appendix A.1).
>
> The two branches are inherently synchronized: each example is fed as a pair of the normalized I/Q time series and its CWT image computed from the same signal segment. In the forward pass, the time-domain encoder and the time–frequency encoder first produce their respective representations, which are then projected to the same dimension, concatenated, and fed jointly into the fusion module for further reasoning.

---

> > ### Author Response · Authors · 2025-11-21
> >
> > Q2:What was the memory footprint of the model and what was the sample size used?
> >
> > Answer:
> >
> > For the final model, both the time-domain encoder and the time–frequency encoder are implemented as stacks of 4 standard Mamba layers, and the fusion module also uses 4 standard Mamba layers with the same configuration. All Mamba layers share the same hyperparameters (state size d_state = 256, model width d_model = 192).This setting has approximately 6.3M trainable parameters, which in 32-bit floating point is on the order of 25 MB of weights.We selected this configuration based on an architecture sweep on the UAV RF dataset, summarized in the table below.As shown, increasing the model size from 0.13M to 6.3M parameters consistently improved performance, reaching 99.92% on seen classes and 83.47% on novel classes (overall 90.13%). Further enlarging the backbone to 18–36M parameters brought no clear gains and even slightly degraded the overall accuracy (87–89%), so we chose the 6.3M configuration as the best trade-off between accuracy and memory footprint.
> >
> > | **Mamba layers** | **d_state** | **d_model** | **Params**   | **Seen (%)** | **Novel (%)** | **All (%)** |
> > |:----------------:|:-----------:|:-----------:|:------------:|:------------:|:-------------:|:-----------:|
> > | 2                | 128         | 24          | 0.134496M    | 76.78        | -             | -           |
> > | 2                | 128         | 48          | 0.311616M    | 82.23        | -             | -           |
> > | 4                | 256         | 96          | 2.4768M      | 96.14        | 79.56         | 85.37       |
> > | **4**            | **256**     | **192**     | **6.336M**   | **99.92**    | **83.47**     | **90.13**   |
> > | 4                | 256         | 384         | 18.2016M     | 98.73        | 81.16         | 87.37       |
> > | 6                | 256         | 384         | 27.3024M     | 98.62        | 82.43         | 88.64       |
> > | 8                | 256         | 384         | 36.4032M     | 99.43        | 82.52         | 89.62       |
> >
> > For the datasets, the UAV RF fingerprint dataset contains about 13k labeled I/Q samples, and the UCI HAR benchmark contains 10,299 time-windowed sequences in total. For pretraining, we use a standard 6:2:2 split into train / validation / test on each dataset.
> >
> > ---
> >
> > Q3:How impactful was normalization on training stability and generalization on your unseen data?
> >
> > Answer:
> >
> > Since our goal is to propose a general training framework rather than a dataset-specific solution, we normalize all I/Q waveforms as the very first step, and apply a consistent log-scale normalization to the CWT branch. This removes trivial gain and power differences between datasets and hardware setups, so the model is not encouraged to exploit dataset-specific amplitude bias and can be more easily transferred to new RF environments.
> >
> > Input normalization is a very standard technique in machine learning to improve optimization stability and generalization , and has also been reported to mitigate power-level bias and improve robustness in RF fingerprinting systems. In our experiments, once this normalization pipeline is used, training remains stable across all datasets with a single learning-rate schedule, and the strong performance on unseen devices and domains is consistent with these prior findings. For this reason we did not include a dedicated ablation on turning normalization on/off, although in our early trials we consistently observed that adding normalization usually brings a small but reliable performance gain.
> >
> > ---
> >
> > Q4:The work is definitely an interesting concept, my worry is that the data was gathered in relatively similar conditions. How can we be certain that it would generalize well on say, open air data vs in-doors data?
> >
> > Answer:
> >
> > To directly address this concern, we further conduct experiments on the subset of WiSig dataset.The subset of WiSig is the ManySig is specifically designed to evaluate receiver- and channel-agnostic RF fingerprinting, ManySig is an officially curated subset from the WiSig authors that removes low-quality recordings, balances the classes, and has been widely adopted as a standard benchmark. It contains signals from 6 transmitters (Tx) and 12 receivers (Rx); each receiver collects 1000 baseband I/Q sequences (each of length is 256) for every transmitter, over four different capture days (2021-03-01, 2021-03-08, 2021-03-15, 2021-03-23). Compared with our controlled UAV setup, ManySig explicitly introduces strong variability in receiver hardware, spatial position and propagation conditions, which directly matches the worry about generalizing across different environments.

---

> > > ### Author Response · Authors · 2025-11-21
> > >
> > > For cross-domain evaluation, we follow the cross-receiver and cross-day protocol of DRIFT (Pan et al., 2025)[1] on the ManySig subset of WiSig. Concretely, we use only the signals from the Source Receivers on Source Day (03-01) to pretrain our framework, then freeze the backbone and train a linear FC head to align transmitter labels for downstream classification. We evaluate on disjoint Target Receivers and Target Days (Day 2–4:03-08,03-15,03-23) as in DRIFT, and additionally compare against Informer[2], a strong architecture for long-sequence classification. As shown in the below table , our PRLS-RFF consistently achieves higher average accuracy than both DRIFT and Informer across all source–target configurations, indicating that the proposed framework maintains superior cross-domain robustness even under large-scale multi-receiver settings and long temporal windows. We will include these ManySig cross-receiver and cross-day results in the main paper in the revised version.
> > >
> > > | Source Receivers      | Source Day | Target Receivers                                  | Target Day | DRIFT (%) | Informer (%) | PRLS-RFF (ours) (%) |
> > > |:------------------------------:|:----------------:|:---------------------------------------------------:|:-----------:|:----------:|:-------------:|:--------------------:|
> > > | (1-1, 1-19, 8-8)     | Day 1      | (14-7, 18-2, 19-2, 2-1, 2-19, 20-1, 3-19, 7-14, 7-7) | Day 2      | 80.37     | 72.41    | **87.42**               |
> > > | (1-1, 1-19, 8-8)     | Day 1      | (14-7, 18-2, 19-2, 2-1, 2-19, 20-1, 3-19, 7-14, 7-7) | Day 3      | 78.05  | 81.33      | **83.57**               |
> > > | (1-1, 1-19, 8-8)     | Day 1      | (14-7, 18-2, 19-2, 2-1, 2-19, 20-1, 3-19, 7-14, 7-7) | Day 4      | 79.38     | 75.41      | **80.43**               |
> > > |                 |            |                                                        | Average    | 79.27    | 76.38       | **83.81**        |
> > > | (1-1, 14-7, 18-2, 7-7)  | Day 1      | (1-19, 18-2, 19-2, 2-1, 2-19, 20-1, 3-19, 7-14)   | Day 2      | 69.70     | 81.14        | **86.41**             |
> > > | (1-1, 14-7, 18-2, 7-7)  | Day 1      | (1-19, 18-2, 19-2, 2-1, 2-19, 20-1, 3-19, 7-14)   | Day 3      | 71.00     | 75.31        | **90.37**             |
> > > | (1-1, 14-7, 18-2, 7-7)  | Day 1      | (1-19, 18-2, 19-2, 2-1, 2-19, 20-1, 3-19, 7-14)   | Day 4      | 68.30     | 64.23        | **83.42**             |
> > > |                     |            |                                                   |   Average  |   69.67 |  73.56     |  **86.73**            |
> > > | (1-1, 1-19, 14-7, 7-7, 8-8)  | Day 1      | (18-2, 19-2, 2-1, 2-19, 20-1, 3-19, 7-14)  | Day 2      | 79.12     | 77.21        | **89.41**               |
> > > | (1-1, 1-19, 14-7, 7-7, 8-8)  | Day 1      | (18-2, 19-2, 2-1, 2-19, 20-1, 3-19, 7-14)   | Day 3      | 82.10     | 81.49        | **83.19**               |
> > > | (1-1, 1-19, 14-7, 7-7, 8-8)  | Day 1      | (18-2, 19-2, 2-1, 2-19, 20-1, 3-19, 7-14)  | Day 4      | 81.64     | 71.14        | **87.43**               |
> > > |                            |            |                                           |  Average  | 80.95     | 76.61      |  **86.68**            |

---

> ### Author Response · Authors · 2025-11-21
>
> For open-world evaluation, we follow the same protocol as in our main paper on the ManySig subset of WiSig. We treat transmitters Tx6-15, Tx8-20, and Tx14-7 as known classes, and transmitters Tx14-10, Tx20-15, and Tx20-19 as unknown classes. The backbone is pretrained only on labeled data from the known transmitters; we then freeze it and train a linear classifier on the known classes, and finally evaluate both closed-set accuracy on seen classes and unknown-class rejection / novel-class discovery on the WiSig open-world split. In addition to k-means, OpenNCD, and the recent open-world RFF baseline Han et al. (2025), we further include Informer as a strong long-sequence backbone. Since Informer is originally designed for closed-set long sequence time-series forecasting rather than open-world recognition, we extend it by applying softmax-thresholding for unknown-class rejection and using DeepDPM, a deep Dirichlet-process mixture clustering method, for novel-class discovery in its feature space. As shown in the below table, our PRLS-RFF achieves the best performance on seen, novel, and overall accuracy (97.43% / 78.48% / 83.47%), outperforming all baselines including Informer and Han et al. (2025), which demonstrates that our framework retains strong open-world robustness on the large-scale, multi-receiver WiSig dataset. We will incorporate these WiSig open-world experiments and results into the main paper in the revised version.
>
> | Method                  | Seen (%) | Novel (%) | All (%) |
> |-------------------------|:--------:|:---------:|:-------:|
> | k-means                 |  43.21   |   37.21   |  40.64  |
> | OpenNCD                 |  87.59   |   65.32   |  74.68  |
> | Informer+DeepDPM      |  94.21   |   43.14   |  61.28  |
> | Baseline (Han et al., 2025) |  92.34   |   62.17   |  73.15  |
> | **PRLS-RFF (ours)**     | **97.43**| **78.48** | **83.47** |
>
>
> **References**
>
> [1] Pan Y, Wang X, Cheng N, et al. Cross-Receiver Generalization for RF Fingerprint Identification via Feature Disentanglement and Adversarial Training[J]. arXiv preprint arXiv:2510.09405, 2025.
>
> [2]Zhou, H.; Zhang, S.; Peng, J.; Zhang, S.; Li, J.; Xiong, H.; and Zhang, W. 2021. Informer: Beyond efficient transformer for long sequence time-series forecasting. In Proceedings of the AAAI conference on artificial intelligence, volume 35, 11106–11115.
>
> ---
>
> Q5:THE UIC HAR dataset is used as proof-of-concept for larger open set identification. How are you handling it? What are the I/Q components here? Is the pipeline the same across both of the datasets?
>
> Answer:
>
> For the UCI HAR dataset, we first reorder and concatenate the 9 channels into a single real-valued sequence and treat it as the I signal. We then apply a discrete Hilbert transform to this sequence to obtain the corresponding quadrature component as the Q signal, which gives us a pseudo-IQ complex representation. Apart from this deterministic input reformatting, the model architecture, training setup, and open-set evaluation protocol for UCI HAR are exactly the same as for the UAV RF dataset, with only the input length and number of classes being different.
>
> ---
>
> Q6:How extensive was hyperparameter tuning? How were the parameters chosen?
>
> Answer:
>
> We ran a series of depth–width sweeps on the UAV dataset and selected the configuration that offers the best accuracy–efficiency trade-off. The results are summarized in the table below. Increasing the backbone from small 2-layer settings to a 4-layer Mamba with d_state = 256 and d_model = 192 substantially improves performance, whereas further enlarging d_model and the number of layers leads to more parameters with only marginal or even negative gains in All accuracy on this benchmark.
>
> | **Mamba layers** | **d_state** | **d_model** | **Params**   | **Seen (%)** | **Novel (%)** | **All (%)** |
> |:----------------:|:-----------:|:-----------:|:------------:|:------------:|:-------------:|:-----------:|
> | 2                | 128         | 24          | 0.134496M    | 76.78        | -             | -           |
> | 2                | 128         | 48          | 0.311616M    | 82.23        | -             | -           |
> | 4                | 256         | 96          | 2.4768M      | 96.14        | 79.56         | 85.37       |
> | **4**            | **256**     | **192**     | **6.336M**   | **99.92**    | **83.47**     | **90.13**   |
> | 4                | 256         | 384         | 18.2016M     | 98.73        | 81.16         | 87.37       |
> | 6                | 256         | 384         | 27.3024M     | 98.62        | 82.43         | 88.64       |
> | 8                | 256         | 384         | 36.4032M     | 99.43        | 82.52         | 89.62       |
>
> Importantly, our framework is backbone-agnostic: the proposed physics-consistent pretraining and dual-stream architecture can be coupled with different Mamba configurations, and the backbone can be re-tuned when transferring to larger or more complex RF datasets.
>
> ---

---

> ### Author Response · Authors · 2025-11-21
>
> Q7:The UC RF dataset is relatively small and in very controlled conditions. How can we be sure that the accuracy presented here can generalize well in open-air, multipath heavy conditions?
>
> Answer:
>
> We have addressed this concern by adding cross-domain and open-world experiments on the WiSig subset ManSig, which is collected in more realistic, multipath-rich conditions. The dataset description, experimental setup, and detailed results are all reported in our response to Question 4; please refer to the answer of question4.
>
> ---
>
> Q8:It would be quite more motivating to see per class confusion breakdowns than simple accuracy, especially in the open-set case.
>
> Answer:
>
> In the open-world experiments, we follow the evaluation protocol of Han et al. and report the standard Seen / Novel / All top-1 accuracies, since their baseline only provides these metrics and we would like to keep the comparison fair.
> To alleviate your concern, we additionally report clustering quality metrics for novel-class discovery (NMI, ARI, and discovery ACC) as logs below:
> UAV:NMI : 0.8146777233232577, ARI: 0.7524811806859501, ACC: 0.83468, current K: 4
> UCIHAR:NMI : 0.8717017634101468, ARI: 0.828020100382347, ACC: 0.92321, current K: 3
>
> ---
>
> Q9:Why is the green cluster on Figure 3b so much more discernible?
>
> Answer:
>
> In Fig. 3b, each color corresponds to a different UAV transmitter. The green class happens to exhibit more consistent and distinctive RF characteristics in this dataset, so its embeddings are more compact and appear as a more clearly separated cluster in the t-SNE plot. The model does not treat this class differently; this is mainly a property of the underlying data distribution and the t-SNE visualization rather than a special case of our method.

---

> > ### Author Response · Authors · 2025-12-02
> > **Summary Comment for AC - Reviewer mELH**
> >
> > Reviewer mELh gave the paper an initial rating of 6 and recognized several key strengths: (i) the physically grounded self-supervised learning scheme based on realistic channel/receiver perturbations, (ii) the use of a Mamba backbone for efficient long-context RF sequence modeling, and (iii) empirically strong domain robustness on the evaluated benchmarks. We sincerely thank the reviewer for this careful and technically insightful assessment, and for the time and effort they invested in evaluating our work.
> >
> > The reviewer’s main concerns were that (a) the UAV dataset alone is small and collected under controlled conditions, so high accuracies might not fully demonstrate real-world robustness; (b) hyperparameter tuning, training cost, and memory footprint of the Mamba backbone were not described in enough detail; (c) UCI HAR, being non-RF, mainly validates open-set mechanics rather than RF-specific invariance.
> >
> > In the revised manuscript, we directly address these points. To go beyond the original UAV setup, we now add extensive experiments on the ManySig subset of the WiSig dataset, which features multiple transmitters, many receivers, and realistic open-air, multipath-rich propagation. On ManySig, we follow DRIFT-style cross-receiver and cross-day protocols and show that PRLS-RFF consistently outperforms both DRIFT and an Informer-based long-sequence baseline across all source–target configurations. We also include open-world experiments on ManySig, where PRLS-RFF achieves higher Seen/Novel/All accuracy than k-means, OpenNCD, Informer+DeepDPM, and the recent open-world RFF baseline we compare against. These additions substantially strengthen our evidence for cross-domain generalization under heterogeneous RF environments and hardware.
> >
> > We further clarify implementation and training details raised in Questions 1–3 and 6–8. The revised paper now explicitly explains synchronization and normalization between the time-domain and time–frequency branches, reports the model size (~6.3M parameters) and memory footprint, and summarizes the depth–width sweeps used to select the final Mamba configuration as an accuracy–efficiency trade-off. We describe how UCI HAR is converted into a pseudo-I/Q representation and emphasize that, apart from this deterministic reformatting, the same pipeline is used across datasets.
> >
> > Overall, we believe that the revised manuscript and our rebuttal fully address all of reviewer mELh’s concerns regarding dataset diversity, generalization, training details, and pipeline clarity. Given the reviewer’s already positive initial rating of 6 and their recognition of the core idea as “a very interesting step towards physics-aware representation learning,” we are confident that the strengthened experimental evidence and clearer presentation in the revised version are likely to earn their full support.

---

### Official Review · Reviewer_j9Bp · 2025-11-03

**Soundness:** 2
**Presentation:** 2
**Contribution:** 2
**Rating:** 2
**Confidence:** 3

**Summary:**

This paper introduces PRLS-RFF, a framework designed to improve the robustness of Radio Frequency Fingerprint Identification (RFFI) against domain shifts (such as changes in channel, receiver distance, and time) and open-world scenarios. The authors propose a dual-stream backbone architecture that utilizes Selective State-Space Models (Mamba) to process and fuse time-domain (I/Q) and time-frequency (Continuous Wavelet Transform - CWT) representations. To train this backbone, the paper employs a self-supervised pretraining approach using a multi-view non-contrastive objective. Crucially, the views are generated using "physics-consistent" augmentations tailored to RF signals, including Additive White Gaussian Noise (AWGN), phase distortion, Carrier Frequency Offset (CFO), and I/Q imbalance. The goal is to learn representations that are invariant to these specific perturbations while retaining device-intrinsic signatures.

Empirical evaluations on a UAV RF fingerprint dataset demonstrate that PRLS-RFF outperforms recent baselines in cross-domain identification (varying distances), few-shot adaptation, and open-world recognition tasks.

**Strengths:**

I appreciate the authors' effort to apply new architectures like Mamba and self-supervised learning to the field of RF Fingerprinting: While dual-stream / dual-encoder architectures and self-supervised learning are established in other fields, their specific application here—coupling Mamba-based long-context modeling with RF-specific physical augmentations for fingerprinting—is new.

**Weaknesses:**

The main weaknesses of the paper are lack of clarity and lack of motivation. A lot of things are presumed, which ultimately weakens the scientific contribution of this paper.

- Architecture design in Section 3.3 and 3.4 seems ad-hoc and not well-justified in the paper. The motivation seems weak. Maybe I missed them, but there seems to be no dedicated ablation / exploration experiments on design choices of the main architecture in Figure 1 and 2 - to name a few, why Mamba block not other blocks (Transformer, CNN, RNN, etc)? how are # parameters decided for each architecture blocks?

- Mamba block in Figure 1 is not clearly introduced in the paper.

- Why the self-supervised loss is "non-contrastive"? I wonder if there is a contrastive counterpart and how it performs.

- The dataset used and the evaluation metric (i.e. top1 accuracy) are not clearly introduced in the paper.

- The baselines in Table one are not clearly described. For example CNN-LSTM is super unclear - there could be numerous ways of designing the architecture and altering the design choices like # parameters each layer, training loss - what we are exactly comparing against here?

- typo: line 366 "about 1 2 points" should be "about 1-2 points"

**Questions:**

Please see weakness for details.

---

> ### Author Response · Authors · 2025-11-21
>
> W1:
>
> Answer:
>
> We thank the reviewer for pointing out that the architectural motivation and backbone choices were not clearly explained in the current version. We clarify our design decisions and add new ablations below.
> **Backbone and the choice of Mamba.**
> Our backbone design is not arbitrary. At both the early and late stages of this work we evaluated several representative long-sequence models from different families. Based on recent literature, we selected Informer[1] (a Transformer variant specifically designed for long time-series forecasting) and Mamba (a selective state-space model that has shown strong performance on long-context sequence modeling), and additionally included strong CNN [2]and RNN[3] backbones for 1D time series. On the UAV dataset, we compared:
>
> | **Backbone**           | **Family**      | **Seen (%)** | **Novel (%)** | **All (%)** |
> |:-----------------------|:---------------:|:------------:|:-------------:|:-----------:|
> | Informer               | Transformer     | 95.47        | 80.37         | 83.62       |
> | **PRLS-RFF (ours)**    | Mamba           | **99.92**    | **83.47**     | **90.13**   |
> | InceptionTime          | CNN             | 78.42        | 64.13         | 71.68       |
> | GRU-FCN                | RNN             | 82.42        | 61.54         | 72.36       |
>
> The results show that our Mamba-based backbone achieves the best overall open-world performance among all four families. This empirically justifies our choice of Mamba as the default backbone, rather than an ad-hoc architectural preference. We will include this ablation table in the appendix.
>
> **Multimodal fusion and mamba blockdesign**
> We use modality-specific encoders for the time-domain and time–frequency (CWT) branches, followed by a shared Mamba fusion stack that operates on the concatenated features. This design is well aligned with the multimodal literature, where unimodal encoders are combined with a shared fusion transformer or gated cross-attention module (similar in spirit to Flamingo-style vision–language fusion)[4], rather than completely separate dual-tower encoders as in CLIP. In our setting, the two "modalities” are deterministic, physics-consistent transforms of the same RF signal, and thus highly correlated and complementary views of the same underlying hardware impairments. Sharing the fusion Mamba parameters encourages cross-view consistency and parameter efficiency, and in practice we found it to be a robust choice。 We will make this motivation more explicit in Sec. 3.3–3.4.
>
> **Mamba backbone depth/width and parameter selection.**
> We did not set the Mamba backbone hyperparameters ad hoc. Instead, we ran a series of depth–width sweeps on the UAV dataset and selected the configuration that offers the best accuracy–efficiency trade-off. The results are summarized in the table below. Increasing the backbone from small 2-layer settings to a 4-layer Mamba with d_state = 256 and d_model = 192 substantially improves performance, whereas further enlarging d_model and the number of layers leads to more parameters with only marginal or even negative gains in All accuracy on this benchmark.
>
> | **Mamba layers** | **d_state** | **d_model** | **Params**   | **Seen (%)** | **Novel (%)** | **All (%)** |
> |:----------------:|:-----------:|:-----------:|:------------:|:------------:|:-------------:|:-----------:|
> | 2                | 128         | 24          | 0.134496M    | 76.78        | -             | -           |
> | 2                | 128         | 48          | 0.311616M    | 82.23        | -             | -           |
> | 4                | 256         | 96          | 2.4768M      | 96.14        | 79.56         | 85.37       |
> | **4**            | **256**     | **192**     | **6.336M**   | **99.92**    | **83.47**     | **90.13**   |
> | 4                | 256         | 384         | 18.2016M     | 98.73        | 81.16         | 87.37       |
> | 6                | 256         | 384         | 27.3024M     | 98.62        | 82.43         | 88.64       |
> | 8                | 256         | 384         | 36.4032M     | 99.43        | 82.52         | 89.62       |
>
>
>
> **Reference:**
>
> [1]Haoyi Zhou, Shanghang Zhang, Jieqi Peng, Shuai Zhang, Jianxin Li, Hui Xiong, and Wancai Zhang. "Informer: Beyond Efficient Transformer for Long Sequence Time-Series Forecasting.” Proceedings of the AAAI Conference on Artificial Intelligence (AAAI), 35(12): 11106–11115, 2021.

---

> ### Author Response · Authors · 2025-11-21
>
> [2]Hassan Ismail Fawaz, Benjamin Lucas, Germain Forestier, Charlotte Pelletier, Daniel F. Schmidt, Jonathan Weber, Geoffrey I. Webb, Lhassane Idoumghar, Pierre-Alain Muller, and François Petitjean. "InceptionTime: Finding AlexNet for Time Series Classification.” Data Mining and Knowledge Discovery, 34(6): 1936–1962, 2020.
>
> [3] Nelly Elsayed, Anthony S. Maida, and Magdy Bayoumi. "Deep Gated Recurrent and Convolutional Network Hybrid Model for Univariate Time Series Classification.” International Journal of Advanced Computer Science and Applications (IJACSA), 10(5): 656–666, 2019.
>
> [4] Jean-Baptiste Alayrac, Jeff Donahue, Pauline Luc, Antoine Miech, Iain Barr, et al. "Flamingo: A Visual Language Model for Few-Shot Learning.” Advances in Neural Information Processing Systems (NeurIPS), 35: 23716–23736, 2022.
>
> ---
>
> W2:Mamba block in Figure 1 is not clearly introduced in the paper.
>
> Answer:
>
> Regarding the internal design of the Mamba blocks, we adopt the standard Mamba architecture from the original work  and do not introduce additional architectural modifications, since the focus of this paper is on the proposed physics-consistent training framework rather than on inventing a new Mamba variant.
>
> ---
>
> W3:Why the self-supervised loss is "non-contrastive"? I wonder if there is a contrastive counterpart and how it performs.
>
> Answer:
>
> Our pretraining follows a BYOL-style self-supervised learning paradigm, which is inherently non-contrastive. Concretely, for each RF signal we construct multiple physics-consistent views (e.g., different noise levels, phase noise, CFO, and I/Q imbalance) and apply an online–target architecture with EMA-updated target encoder, as in BYOL[1].The self-supervised loss only aligns the representations of these multiple views of the same signal (online predictions vs. target embeddings), and does not use any explicit negatives .In contrast, SimCLR[2] and MoCo [3]are based on a contrastive objective: two augmented views of the same instance form a positive pair, while all other instances in the batch are treated as negatives, and the backbone is explicitly encouraged to push different instances far apart in the feature space.
>
> We deliberately adopt the BYOL-style non-contrastive formulation because it is better aligned with the goal of our work: learning RF fingerprints that are primarily driven by hardware-consistent characteristics, rather than by channel or environmental artifacts. In realistic RF settings, different devices are often observed under varying channels, propagation conditions, and SNRs. A strong contrastive instance-discrimination loss can easily exploit these channel-related differences as shortcuts to separate instances, instead of focusing on device-intrinsic impairments. By only aligning multiple physics-consistent views of the same signal and never repelling other samples, the BYOL-style objective restricts each example to see only its own views and thus helps suppress non-hardware factors in the learned representations.
>
> We additionally implement a contrastive counterpart of our method on the UAV dataset, where we introduce a standard negative-sample-based contrastive loss between different signals while keeping the rest of the pretraining pipeline unchanged. We train all loss variants with the same backbone, physics-consistent augmentations, and pretraining hyperparameters as in the main paper, and report their open-world identification performance.
>
> | **Loss variant**                        | **Seen (%)** | **Novel (%)** | **All (%)** |
> |:---------------------------------------|:------------:|:-------------:|:-----------:|
> | BYOL-style non-contrastive loss        | **99.92**    | **83.47**     | **90.13**   |
> | SimCLR-style contrastive loss          |   99.14      |    71.56     |   82.46     |
>
>
> **Refrence**
>
> [1]Grill J B, Strub F, AltchÃ© F, et al. Bootstrap Your Own Latent: A new approach to self-supervised learning[C]//Neural Information Processing Systems. 2020.
>
> [2]Chen T, Kornblith S, Norouzi M, et al. A simple framework for contrastive learning of visual representations[C]//International conference on machine learning. PmLR, 2020: 1597-1607.
>
> [3]K. He, H. Fan, Y. Wu, S. Xie and R. Girshick, "Momentum Contrast for Unsupervised Visual Representation Learning," 2020 IEEE/CVF Conference on Computer Vision and Pattern Recognition (CVPR)

---

> ### Author Response · Authors · 2025-11-21
>
> w4:The dataset used and the evaluation metric (i.e. top1 accuracy) are not clearly introduced in the paper.
>
> Answer:
>
> We thank the reviewer for pointing out this clarity issue. Our experiments use two public benchmarks。
> **UAV RF fingerprint dataset** for cross-domain identification and open-world recognition. As described in Sec. 4.1, this dataset contains I/Q signals from seven identical DJI M100 UAVs collected at four receiver distances (6, 9, 12, 15 ft) and four time-separated acquisition bursts per distance, inducing both distance and session shifts.
> **UCIHAR** for open-world recognition only, following . We use its six activity classes and split them into three seen and three novel classes.
> Regarding the evaluation metrics.
> For cross-domain identification (Sec. 4.1), we follow Cai et al. (2024) [1] and report top-1 classification accuracy, i.e., the percentage of test examples whose predicted label matches the ground-truth label, averaged over the four leave-one-subset-out experiments on data#1 and data#2 (see Table 1 and Table 2).
>
> For open-world recognition (Sec. 4.2), we follow Han et al. (2025) [2] and explicitly report three metrics: Seen, Novel, and All. As stated in Sec. 4.2, Seen is top-1 accuracy on known classes, Novel is discovery accuracy on the rejected unknowns after cluster-to-class matching, and All is overall accuracy on the union of seen and novel samples (Tables 3–6).
> We agree that this information is currently distributed across different parts of Sec. 4, which may make it less visible on a first read. We will add a short paragraph at the beginning of Sec. 4 to clearly present the datasets and metrics used in our experiments.
>
> **Refrence**
>
> [1]Zhenxin Cai, Yu Wang, Guan Gui, and Jin Sha. Toward robust radio frequency fingerprint identification via adaptive semantic augmentation. IEEE Transactions on Information Forensics and Security, 2024.
>
> [2]Zehua Han, Jing Xiao, Qirui Zhao, Zhexuan Cui, Yufeng Wang, Duona Zhang, and Wenrui Ding. Open-world radio frequency fingerprint identification via augmented semi-supervised learning. In Proceedings of the AAAI Conference on Artificial Intelligence, volume 39, pp. 264–272, 2025.
>
> ---
>
> W5:The baselines in Table one are not clearly described. For example CNN-LSTM is super unclear - there could be numerous ways of designing the architecture and altering the design choices like # parameters each layer, training loss - what we are exactly comparing against here?
>
> Answer:
>
> We apologize for not clearly indicating the references and sources in the current draft. The compared models (DNN, CNN-LSTM, and GRU) and their reported results in Table 1 are directly taken from the baseline work we follow Cai et al. (2024) [1]. The detailed architectures of these models are specified in their original papers [Jafari et al., 2022[2]; Cai et al., 2022[3]; Shen et al., 2022[4]]. In the revised version, we will explicitly add the proper citations in the experimental section to clarify that these baselines and their results are inherited from the above works.
>
> **Refrence**
>
> [1]Zhenxin Cai, Yu Wang, Guan Gui, and Jin Sha. Toward robust radio frequency fingerprint identification via adaptive semantic augmentation. IEEE Transactions on Information Forensics and Security, 2024.
>
> [2]]H. Jafari et al., "Signature-aware RF exploitation (SNARE) fingerprinting using deep learning to identify UAVs,” IEEE Aerospace Conference, 2022.
>
> [3]Z. Cai, Z. Liu, and L. Kou, "Reliable UAV monitoring system using deep learning approaches,” IEEE Trans. Rel., vol. 71, no. 2, pp. 973–983, Jun. 2022.
>
> [4]G. Shen, J. Zhang, A. Marshall, and J. R. Cavallaro, "Towards scalable and channel-robust radio frequency fingerprint identification for LoRa,” IEEE Trans. Inf. Forensics Security, vol. 17, pp. 774–787, 2022.
>
> ---
>
> W6:typo: line 366 "about 1 2 points" should be "about 1-2 points"
>
> Answer:
>
> We thank the reviewer for pointing out this typo. We will carefully proofread the manuscript and correct this and other typographical errors in the revised version.

---

> > ### Author Response · Authors · 2025-12-02
> > **Summary Comment for AC - Reviewer j9Bp03**
> >
> > Reviewer j9Bp03 acknowledged the novelty of bringing a Mamba-based dual-stream backbone and physics-consistent multi-view self-supervised pretraining into RF fingerprinting, and recognized that our method improves robustness to domain shifts and open-world scenarios. Their main concern was that the backbone architecture and design choices in Sec. 3.3–3.4 appeared ad hoc and insufficiently motivated, with limited ablations. They also requested clearer explanations of the non-contrastive loss, the UAV dataset and evaluation metrics, the baseline models, and pointed out minor typos.
> >
> > In the rebuttal and revised paper, we substantially clarified the backbone design and added new experiments. Sec. 3.3 now gives a more explicit description of the dual-stream Mamba blocks and fusion strategy, and Sec. 4 clearly states that all experiments use a 4-layer Mamba configuration with state dimension 256 and model width 192. Appendix A.1 reports new ablations across backbone architecture (Informer, Mamba, CNN, RNN) and across Mamba depth/width settings, showing that the chosen Mamba backbone provides the best accuracy–efficiency trade-off and is therefore not an ad-hoc choice.
> >
> > To address the question about the “non-contrastive” loss, Sec. 3.4 and Appendix A.2 now more clearly describe the BYOL-style objective and compare it to a SimCLR-style contrastive variant under the same framework. Using identical backbones, augmentations, and training schedules, the non-contrastive loss achieves similar Seen accuracy but significantly higher Novel and All open-world performance on the UAV dataset, which empirically supports our design choice.
> >
> > At the beginning of Sec. 4 (Experiments), we now provide a concise overview of the UAV RF fingerprint dataset, the WiSig (ManySig) subset, and UCIHAR, and we explicitly define the Top-1, Seen, Novel, and All metrics used in our evaluation. Tables 1–6 more clearly separate cross-domain identification and open-world recognition results. In addition, we clarified that the CNN-LSTM, GRU, and other baselines and their reported numbers are taken from prior work, and we added the corresponding citations in the experimental section. We also fixed the typo the reviewer pointed out and conducted a broader proofreading pass.
> >
> > Overall, we believe that the revised manuscript and our rebuttal fully address all of reviewer j9Bp03’s concerns about architectural motivation, backbone selection, and experimental clarity. Although this reviewer did not post a follow-up after the rebuttal, we are confident that the strengthened motivation, added ablations, and clearer experimental protocol should resolve their earlier reservations about soundness and contribution and are likely to earn their support.

---

### Author Response · Authors · 2025-12-01

Dear AC,

This comment is to help you quickly see what has changed in the revised (rebuttal) version of our manuscript and how these changes address the reviewers’ comments. All modifications are highlighted in the uploaded PDF for easier reading.

Below we summarize the main edits:

1. **Self-supervised loss and “positives-only” design (Reviewer j9Bp – Weakness 3)**
   In **Sec. 3.4**, we expanded the discussion around our multi-view, non-contrastive self-supervised loss (the sentence ending with “…so each signal only interacts with its own views…”). We now explicitly explain why a positives-only, BYOL-style formulation is beneficial in our RF setting, and how it biases the representation toward hardware-consistent fingerprints rather than domain artifacts. We also added a new ablation in **Appendix A.2** comparing our BYOL-style loss with a SimCLR-style contrastive variant, and report that the non-contrastive loss yields better Novel/All performance in the open-world setting.

2. **Backbone configuration, model size, and Mamba hyperparameters
   (Reviewer j9Bp – Weakness 1; Reviewer mELh – Question 2; Reviewer 9KV6 – Weakness 4)**
   In **Sec. 4**, we clarified the implementation of the Mamba blocks used in all experiments (“…implemented as stacks of four standard Mamba layers with state dimension d_state = 256 and d_model = 192…"). We now explicitly state that this configuration was chosen as a favorable accuracy–efficiency trade-off based on ablations. The corresponding backbone-family comparison and Mamba hyperparameter ablations (including parameter counts and memory footprint implications) are summarized in the main text and reported in detail in **Appendix A.1.1–A.1.2**. This addresses questions about how the Mamba parameters were decided, what model sizes are used, and the relation between depth/width and performance.

3. **More challenging cross-domain experiments on WiSig / ManySig
   (Reviewer mELh – Weakness 1 & 3, Question 4, Reviewer rVm4 – Question 2, Reviewer 9KV6 – cross-receiver concern)**
   In **Sec. 4.1**, we extended the cross-domain evaluation beyond the UAV dataset by adding experiments on the **ManySig subset of the WiSig dataset**, which introduces both cross-receiver and cross-week domain shifts. We follow the DRIFT protocol and report these results in **Table 3**, with analysis in the same section. This directly addresses concerns that our cross-domain setting was not sufficiently complex and that cross-receiver, longer-term temporal shifts were missing. At the same time, we introduce **Informer** as a strong Transformer-based baseline in this cross-domain setting, responding to Reviewer 9KV6’s request for stronger baselines.

4. **Open-world WiSig experiments, t-SNE visualization, and strong baselines
   (Reviewer 9KV6 – strong baseline concern)**
   In **Sec. 4.2**, we added the ManySig (WiSig) dataset to the open-world experiments and specified the seen/novel transmitter split. We updated **Figure 3** to include t-SNE visualizations of the latent representation for WiSig, alongside UCIHAR and UAV. We also added **Table 6**, which reports open-world Seen/Novel/All performance on ManySig, including **Informer+DeepDPM** as a strong baseline. These additions strengthen the open-world evaluation and clarify that our method improves over strong sequence models in a challenging cross-receiver, cross-time domain.

5. **Backbone architecture choice and Mamba hyperparameter ablations
   (Reviewer j9Bp – concerns about model design and parameter choices)**
   We added a dedicated set of ablations in **Appendix A.1**, covering:
   - Backbone family comparison (Informer, InceptionTime, GRU-FCN, Mamba) under the same PRLS-RFF framework (**Table 9**)
   - Mamba hyperparameter ablations over number of layers, state dimension, and model width, with parameter counts and open-world metrics (**Table 10**).

   The main conclusions from these ablations (why we adopt Mamba and why we choose the 4-layer, \(d_{\text{state}} = 256\), \(d_{\text{model}} = 192\) configuration) are summarized in the main text, while detailed settings and results are in the appendix due to space limits.

6. **Preprocessing for the non-RFF UCI HAR dataset
   (Reviewer mELh – question on handling non-RF data)**
   In **Appendix A.3**, we describe how we adapt UCI HAR to our complex-valued PRLS-RFF pipeline. We explain that we reorder and concatenate the 9 real-valued sensor channels into a single real sequence (treated as the I component), then apply a discrete Hilbert transform to obtain the Q component and form a pseudo-IQ complex sequence.

We hope this summary helps you quickly navigate the revised manuscript and see how the new experiments and clarifications address the reviewers’ main concerns.

Thank you very much for your time and effort in handling our paper.

---

### Meta-Review · Area_Chair_hgLd · 2026-01-06

**Summary:**

The paper presents a Mamba-based architecture for radio-fingerprinting, evaluated over several public datasets.

The reviewers pointed several weaknesses, some of which are important and, in my opinion, they are not sufficiently addressed in the rebuttal.

1) The paper falls below the bar for reproducibility and fair comparison to cometitors. As pointed out by  Reviewer j9Bp, it is not clear if any hyperparameter exploration was done on competitors (in fact, it seems from the author response that no exploration was done, but topologies/hyperparameters were taken verbatim from other papers).

2) The methodological contribution is incremental and the solution is heuristic/ad hoc (j9Bp,rVm4,9KV6).

3) It is not clear if competitors are state of the art (9KV6).

4) Experiments largely consider limited dataset diversity and scale, bringing generalization into question (9KV6, meLH).

**Reviewer Concerns:**

My recommendation is to reject the paper. Even though concern 4 is partially addressed by additional experiments conducted by the authors, other concerns remain post-rebuttal. In particular, regarding 1-3:

1) Reproducibility issues should not be resolved at the camera-ready stage; this is sine-qua-non for ML papers. It is also concerning that the rebuttal indicates that competitor methods were taken verbatim from other papers with out any hyperparameter exploration; this does not seem to be an apples to apples comparison.

2) In the end, this is an "apply an ML algorithm (Mamba) to a downstream task" paper. I agree with the reviewers that the methodological contribution would have been strenghtened by exploring new methods that indeed incorporate the physical aspects of fingerprints, as opposed to applying yet another ML algorithm to this problem.

3) I too am not convinced that comparison is to SotA methods. A simple search for "machine learning + radio fingerprinting" returns many publications not mentioned by the authors. There is also an issue of scope. From the fact that most of citations in the paper (as well as the ones returned by the search) are in communications/networking venues, and that the contribution lies there, rather than in ML, it seems more appropriate to target such a venue with this work: it is possible that it would then be placed in the right context.

**Reviewer Scores:**

Review scores were: reject, marginally above, reject, marginally above. I think only the last reviewer appears to have been swayed. As methodological contribution concerns raised were not addressed, and the reproducibility/hyperparameter exploration issue remains, I do not expect other reviewers to have change their mind.

---

### Decision · Program_Chairs · 2026-01-26

Reject